# Neural codes track prior events in a narrative and predict subsequent memory for details
Silvy H. P. Collin [1] ✉, Ross P. Kempner[2], Sunita Srivatsan[2] & Kenneth A. Norman [2,3] ✉

Throughout our lives, we learn schemas that specify what types of events to expect in particular contexts and the temporal order in which these events usually occur. Here, our first goal was to investigate how such context-dependent temporal structures are represented in the brain during processing of temporally extended events. To accomplish this, we ran a 2-day fMRI study ($N = 40$) in which we exposed participants to many unique animated videos of weddings composed of sequences of rituals; each sequence originated from one of two fictional cultures (North and South), where rituals were shared across cultures, but the transition structure between these rituals differed across cultures. The results, obtained using representational similarity analysis, revealed that context-dependent temporal structure is represented in multiple ways in parallel, including distinct neural representations for the culture, for particular sequences, and for past and current events within the sequence. Our second goal was to test the hypothesis that neural schema representations scaffold memory for specific details. In keeping with this hypothesis, we found that the strength of the neural representation of the North/South schema for a particular wedding predicted subsequent episodic memory for the details of that wedding.

By integrating across multiple experiences, we can learn about regularities in how events are sequenced in a particular situation (e.g., going to a movie), and we can leverage this knowledge to make predictions about what will happen next; this knowledge about event structure is typically referred to as an event schema or script[1,2]. In the real world, the structure of event sequences is often history-dependent (i.e., non-Markovian), meaning that current sensory observations are not sufficient to determine what will happen next; rather, we have to combine our current sensory input with a representation of what happened in the recent past to make an accurate prediction about the future. For example, consider a situation where you observe a character in a movie standing by the door of their apartment. This observation, on its own, is compatible with a very wide range of next scenes (the character could be going into the apartment, leaving for another destination, etc). However, if you remember that—in the previous scene—the character was sitting in their apartment discussing how they want to get food, this substantially narrows down the space of possible next scenes.

How exactly does the viewer in this example use the past to predict the future? Importantly, there are many distinct ways that a person could leverage the information from the previous scene. One possibility is that the viewer could load up a representation of a *specific sequence ("path") of actions* (e.g., if, in the past, they've repeatedly seen this character walk to their favorite hot dog stand when they are hungry). Another possibility is that the viewer could load up an *abstract schema*—a situation-specific predictive model for how the character will behave, not tied to a particular sequence (e.g., "if the character is hungry, they will be more likely to go into restaurants they encounter instead of walking by them, and they will be irritable when encountering people because they've acted this way when hungry in the past"[3]). A third possibility is that the viewer could simply carry forward a representation of the previous scene, without loading into memory a specific predicted sequence or abstract schema. This could be useful in several ways (e.g., if you know that the character was just in their apartment and you see them outside the apartment, you can predict that they're in the process of leaving the apartment, rather than coming back to it). A computational challenge inherent in this "carrying forward" strategy is that, if the viewer uses the exact same representation for when a scene is *presently happening* vs. *happened in the past*, this could lead to confusion about what is presently happening; as we will discuss below, one possible way to disambiguate representations of past and present events is to rotate representations of past events in the brain[4–7].

[1]Tilburg School of Humanities and Digital Sciences, Tilburg University, Tilburg, Netherlands. [2]Princeton Neuroscience Institute, Princeton University, Princeton, USA. [3]Department of Psychology, Princeton University, Princeton, USA. ✉e-mail: s.h.p.collin@tilburguniversity.edu; knorman@princeton.edu

Previous fMRI studies have identified history-dependent representations in the brain during context-dependent sequence learning tasks[8,9] but these studies have not sought to discriminate between the three possibilities listed above. The *first goal* of our study is to test for all three types of history-dependent representation during context-dependent sequence learning. The *second goal* of our study is to look at how these different types of history-dependent representation affect subsequent memory for episodic details. A large body of prior work has demonstrated that schemas can support encoding of specific event details[10–15], and other work has argued that the inferred schema serves to contextualize episodic memories[16,17]—e.g., memories formed while your "looking for food" schema was active will be easier to access when that same schema is active later. In keeping with this claim, a recent study found that the strength of neural schema representations during perception of audio and movie narratives was related to later memory for details from these narratives[15]. However, what remains to be investigated is whether it is indeed specifically the schema representations that relate to memory for details, rather than the other types of history-dependent memory representations described above. As described below, the design of our study makes it possible to investigate the possible relation between later memory for episodic details and several types of history-dependent memory representations (schema representation, path representation, rotated representation of preceding rituals) during encoding.

To achieve these goals, we ran a 2-day functional magnetic resonance imaging (fMRI) study (Fig. 1) in which we exposed participants to many unique animated videos of wedding ceremonies with sequences of rituals originating from a novel (fictional) culture, where each video was accompanied by a unique narrative. The rituals unfolded according to the graph structure shown in Fig. 1B. Each wedding was from one of two fictional cultures (indicated by the couple being from the North or the South of a fictional Island). At the start of each wedding video, the culture (North or South) was mentioned (but this information did not persist on screen, so participants had to remember it). After a short introduction to the wedding,

one of two "stage 2" rituals (campfire or flower) was shown, with equal probability—there was no way to predict which stage 2 ritual would be shown. Following this unpredictable stage 2 ritual, the subsequent rituals were fully predictable given knowledge of whether the wedding was for a couple from the North or South (e.g., in North weddings, campfire was always followed by coin, and flower was always followed by torch). Importantly, while the weddings from the two cultures incorporated the same rituals (with the exact same frequency), the *transition structures* of the weddings were distinct and opposite from each other (e.g., in South weddings, campfire was followed by torch, not coin; and flower was followed by coin, not torch). This sequence design leads to the non-Markovian property that (starting with the transition from stage 2 to stage 3) participants need to represent what happened earlier in the wedding in order to predict what rituals they will observe going forward. In a previous study that measured behavior only (i.e., no fMRI), we demonstrated using these materials that participants could successfully learn the ritual transition structure of the weddings through experience (i.e., without seeing the actual graph shown in Fig. 1B) if they were given blocked training (i.e., where they viewed multiple weddings from the North, followed by multiple weddings from the South), but they performed much worse when given interleaved training (i.e., strictly alternating weddings from the North and South[18]). Here, we leveraged this blocked training procedure on day 1 of our study to facilitate successful learning of the ritual transition structure. Next, participants came back on day 2 to view ceremonies that also followed this structure. An important feature of the present study is that each wedding video contained details unique to that wedding; on day 2, we tested participants on their memory for these unique details after they viewed the weddings—this arrangement allowed us to assess the relationship between history-dependent neural representations in the brain and subsequent memory for wedding-specific details.

An important feature of our task design is that it gave us the ability to tease apart the three kinds of history-dependent neural representations described above:

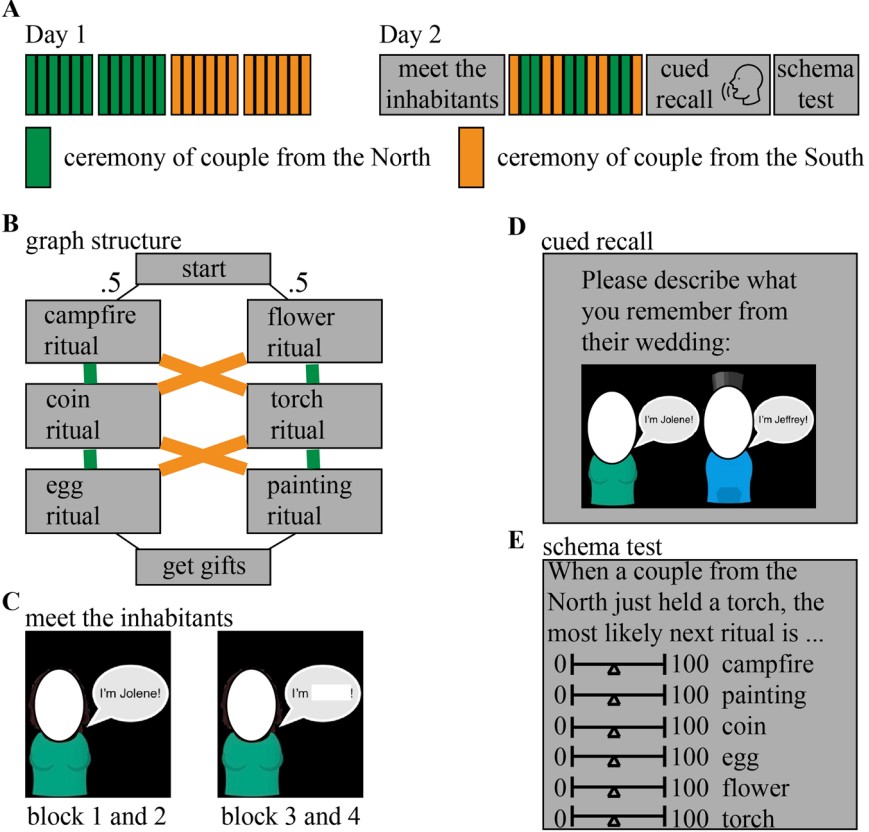

**Fig. 1 | Task design. A** Overview of the experiment (images generated with Unity, https://unity3d.com/), consisting of two days. **B** Graph structure that indicates how the ritual sequences unfold in the North graph vs the South graph. The critical sections of this graph structure for analyses are stages 2 (campfire, flower), 3 (coin, torch) and 4 (egg, painting). **C** Screenshots of the pretraining task in which participants were introduced to the characters before viewing the actual wedding videos. Pictures of faces omitted in the figure for privacy reasons. **D** Screenshot of the (self-paced) cued recall task that participants performed on day 2 of the experiment. Pictures of faces omitted in the figure for privacy reasons. **E** Screenshot of the schema test.

- Abstract schema code (North / South): If participants express one neural pattern throughout North weddings that is distinct from the neural pattern expressed throughout South weddings, this is "abstract" in the sense that it can not be explained in terms of representing specific rituals (on average, each ritual occurs equally often in North and South weddings) or in terms of representing a single sequence (since North and South each encompass multiple sequences of rituals; e.g., North weddings can follow the campfire, coin, egg sequence or the flower, torch, painting sequence). As noted above, this representation (if present) would be sufficient to fully predict all of the rituals that are predictable (i.e., in stages 3 and 4). We predicted this schema code to be present in a network of brain regions identified as schema-sensitive regions in a similar study by Baldassano, Hasson, and Norman[19].

- Path (sequence) code: In addition to (or instead of) representing the North / South schema, participants could represent the specific "path" (sequence of rituals) that a particular wedding follows. Each of the North and South schemas encompasses two distinct paths, making for four possible paths in total. Knowing the schema plus the identity of the current ritual makes it possible to deduce the full path (e.g., North + campfire uniquely specifies that the full path will be campfire, torch, egg). Path codes can be viewed as instances of a schema or script (in the sense that they capture a regularity that is present across multiple observed weddings), but they are less abstract than a representation of North / South, which (as noted above) would encompass multiple paths. Path codes are also somewhat less useful for prediction than North / South codes (insofar as they can not be used to predict the stage 2 to 3 transition without also knowing whether the couple was from the North or South). We predicted this path code to be present in the hippocampus since the hippocampus is known to represent predictable sequences[8,9].

- Preceding ritual code: In addition to the above codes, participants could also represent the immediately preceding ritual. On its own, this would not be useful for predicting the stage 2 to 3 transition (i.e., simply knowing that the preceding ritual was campfire, without knowledge of whether the couple was from the North or South, does not specify whether the next ritual will be torch or coin). However, accurately representing the current stage 3 ritual and the preceding (stage 2) ritual would be sufficient to predict the stage 4 ritual (i.e., knowing that torch was preceded by campfire is sufficient to deduce that the next ritual will be egg). As noted above, representing the preceding ritual and the current ritual at the same time could cause confusion if both are represented in the same form. A recent study looking at context-dependent sequence representation in mice[7] suggests that one can avoid confusion between current and preceding items by rotating the neural code for the preceding item (note that, here, we are using "rotation" in the linear algebraic sense). fMRI and EEG studies of working memory tasks in humans have obtained evidence consistent with this view, suggesting that rotations are used to differentiate between currently relevant and currently irrelevant items held in working memory[4–6]; these human studies specifically found that representations of currently irrelevant items were represented in a format that is anticorrelated with the representations of currently relevant items, analogous to a "photo negative" (i.e., such that neural features that are especially active when an item is currently relevant are especially inactive when that same item is currently irrelevant). Based on these findings, we expected that visual cortex representations of preceding rituals, if present, would be anticorrelated with the patterns that are evoked by these same rituals when they are currently happening. We do not expect such anticorrelated patterns for schema or path codes, since, unlike for rituals, there is no need to represent multiple schemas or paths at the same time in order to accurately predict upcoming events.

In summary, there were (at least) three distinct ways that participants in our study could represent the context-dependent transition structure of the wedding videos: an abstract North/South schema code (henceforth, *schema code*); a representation of specific sequences (henceforth, *path code*); and a representation of the preceding ritual, which we hypothesized would be rotated (henceforth, *rotated preceding ritual code*). We also expected that some regions would simply represent the current ritual (henceforth, *current ritual code*).

## Methods

There is no preregistration for this study.

### Participants

Forty-four participants with normal or corrected-to-normal vision and no hearing impairments participated in this 2-day fMRI study. Four participants did not finish the entire study. Therefore, the final sample consisted of 40 participants (11 men, 29 women, ages 18–35, mean age 21.4). All participants were either native English speakers or started speaking English before the age of seven. Two participants were left-handed. All participants provided written informed consent before the start of the study in accordance with experimental procedures approved by the Princeton University Institutional Review Board (IRB protocol 7883). Participants received monetary compensation for their time (20 USD per hour, and additionally a performance bonus of 10 USD when completing both days of the experiment). The study consisted of two sessions on two consecutive days (both 2 h long). No statistical methods were used to predetermine the sample size, but our sample size is similar to those reported in similar publications of neuroimaging schema experiments (e.g., see refs. 19,20, both of which tested between 30 and 40 participants).

### Materials

We exposed participants to many unique animated videos of wedding ceremonies (generated with Unity, https://unity3d.com/) with fictional rituals originating from a novel (fictional) culture, where each unique video was accompanied by a unique (audio) narrative. The audio was imported into Audacity for further processing before it was integrated with the video. The audio files were imported together, and then normalized. The remaining audio processing was done in iMovie: the auto sound editor, background noise reducer, and equalizer ("voice enhance") were used. Furthermore, the volume of each video was adjusted to a subjectively good level. Each of these ceremonies had a duration of approximately 2 min and was composed of a sequence of rituals. Participants had to discover how the sequence of these stages in a ceremony could be predicted based on the context (crucially, participants were not shown the ground-truth transition structure in Fig. 1 or directly informed that the transition structure depended on the couple being from the North vs. South; they had to discover these contextual dependencies on their own by watching the weddings). Each ceremony, despite the regularities in the type and order of the rituals, was unique with respect to details (e.g., the characters, objects, and audio narrative).

When a wedding couple came from the North, the rituals performed during their wedding followed one of the following two paths:
- North (path-A): start of the wedding—celebrate around campfire—drop coin in bowl—break an egg—receive gifts
- North (path-B): start of the wedding—plant a flower—hold a torch—draw a painting—receive gifts

When a couple came from the South, the rituals performed during their wedding followed one of the following two paths:
- South (path-C): start of the wedding—celebrate around campfire—hold a torch—break an egg—receive gifts
- South (path-D): start of the wedding—plant a flower—drop coin in bowl—draw a painting—receive gifts

### Experimental procedure

**Day 1**. Participants were told that they would be watching animated videos that depict rituals performed during wedding ceremonies from

couples that all live on a certain island. Additionally, they were told that some couples live in the North part of the island and some in the South part of the island, but that they all traveled to the city Navahla for their wedding. They were informed that it is their task to learn to predict which rituals are more likely to happen next. The videos were projected using an LCD projector onto a rear-projection screen located in the magnet bore and viewed with an angled mirror. PsychoPy was used to display the task. Audio was delivered using in-head earphones.

Learning. Participants were exposed to 24 ceremonies (12 of each context) in the following order: two blocks of the first context (either North or South, counterbalanced across participants) of six ceremonies each, followed by two blocks of the other context with six ceremonies each. The videos were presented continuously. There was a break of 2 min after each block. To assess their learning of the four paths, they received two-alternative forced choice questions asking them to predict what would happen next. They received these questions during the first and last weddings from each of these four blocks. Within these weddings, the questions were presented right before the start of stage three (drop coin in bowl versus hold a torch) and right before the start of stage four (break an egg versus draw a painting). Participants had a fixed time window of 4 s to answer these prediction questions. Additionally, to keep the participants engaged with the task continuously throughout the experiment, they received six two-alternative forced choice questions after each wedding (the presentation time was 2 s per question, no ITI) asking which episodic detail (of two random episodic details presented) they saw during the preceding wedding. They gave their answer using a button box. We monitored people's alertness throughout the task via an MRI-compatible eye-tracking camera. fMRI data from day 1 are not reported in this paper.

## Day 2
Pretraining ("meet the inhabitants"). Before the start of the MRI session of day two, participants performed one task outside of the scanner behind a computer screen. During this task, they were introduced to new characters (different from the ones they saw on the first day). They were instructed that they would again see a number of ceremonies today and that they would first meet the inhabitants of the island whose ceremonies they would be viewing, before seeing the actual ceremonies. They received a picture of each character in isolation (24 characters in total). First, they received two blocks in which they saw all 24 characters along with their names, one at a time, and were told to try to remember their names. Next, they received two blocks in which they were only presented with pictures of the characters, again one at a time, and were told to write the person's name in the answer field. They received feedback about whether they were correct or not. If they were incorrect, the feedback stated the correct name.

Watching novel ceremonies. After the pretraining, participants moved on to the MRI-session of that day. Participants were exposed to a novel set of six North and six South ceremonies in interleaved order (i.e., SNNSSNNSSNNS). The videos were presented continuously. While watching, participants again received two-alternative forced choice questions asking to predict what would happen next during the first and last ceremony they saw; these questions were presented right before the start of stage three (drop coin in bowl versus hold a torch) and right before the start of stage four (break an egg versus draw a painting). Participants had a fixed time window of 4 s to answer these prediction questions. They again were given the six general attention questions at the end of each ceremony to keep them engaged with the ceremonies.

Cued recall test. After they completed watching the novel ceremonies, participants performed a spoken recall session in the MRI-scanner. During this recall session, participants received pictures of the wedding couples and their names, one at a time, as a cue. They were instructed to describe anything they recalled from that couple's wedding. They were specifically instructed that any detail they remembered would be important to mention.

Participants were instructed to indicate verbally when they were finished, after which the experimenter would move on to the next couple. Participants' speech was recorded using a customized MR-compatible recording system. Both watching the novel ceremonies as well as the recall test were displayed using PsychoPy. The fMRI data associated with this recall test was not taken into consideration for the purpose of this article.

Test for schema learning. Outside of the MRI scanner, we tested their memory for the ritual transitions during each of the four paths. They received questions like, "If a couple from the North just planted a flower, what is the most likely ritual to happen next?". Participants were given all possible answer options (i.e., all possible rituals) and were told to allocate a total of 100 percent across these answer options, explaining that they were free to either put the entire 100 percent on one answer or split it across multiple answers. They received eight questions:
- North couple, right after celebrating around a campfire,
- North couple, right after planting a flower,
- North couple, right after dropping a coin in a bowl,
- North couple, right after holding a torch,
- South couple, right after celebrating around a campfire,
- South couple, right after planting a flower,
- South couple, right after dropping a coin in a bowl,
- South couple, right after holding a torch

## MRI acquisition
fMRI data were collected on a 3T Prisma scanner (Siemens). Functional data was acquired using a whole-brain multiband EPI sequence, TR = 1.5 s, TE = 30 ms, voxel size 2 mm isotrophic, multiband factor = 3. We collected two anatomical scans (MPRAGE), one on each day, voxel size 1 mm isotrophic, TR = 2300 ms, TE = 2.96 ms. To correct for distortions, we collected two fieldmaps on each day; one in the anterior-posterior phase encoding direction and one in the posterior-anterior phase encoding direction, TR = 8000 ms, TE = 66 ms, voxel size 2 mm isotrophic. Additionally, we collected a T2 turbo-spin-echo scan (to create subject-specific hippocampal subfields masks using ASHS, https://www.nitrc.org/projects/ashs).

## MRI data preprocessing
Results included in this manuscript come from preprocessing performed using fMRIPprep 1.2.3[21,22], which is based on Nipype 1.1.6-dev[23,24].

**Anatomical data preprocessing.** There were 2 T1-weighted (T1w) images acquired (one on each day). Both were corrected for intensity non-uniformity (INU) using N4BiasFieldCorrection (ref. 25, ANTs 2.2.0). A T1w-reference map was computed after registration of 2 T1w images (after INU-correction) using mri robust template (FreeSurfer 6.0.1[26]). The T1w-reference was then skull-stripped using antsBrainExtraction.sh (ANTs 2.2.0), using OASIS as target template. Brain surfaces were reconstructed using recon-all (FreeSurfer 6.0.1[27]), and the brain mask estimated previously was refined with a custom variation of the method to reconcile ANTs-derived and FreeSurfer-derived segmentations of the cortical gray-matter of Mindboggle[28]. Spatial normalization to the ICBM 152 Nonlinear Asymmetrical template version 2009c[29] was performed through nonlinear registration with antsRegistration (ANTs 2.2.0[30]), using brain-extracted versions of both T1w volume and template. Brain tissue segmentation of cerebrospinal fluid (CSF), white-matter (WM) and gray-matter (GM) was performed on the brain-extracted T1w using fast (FSL 5.0.9[31]).

**Functional data preprocessing.** For each of the 6 BOLD runs found per subject (day1-run1, day1-run2, day1-run3, day1-run4, day2-video watching, day2-recall), the following preprocessing was performed. First, a reference volume and its skull-stripped version were generated using a custom methodology of fMRIPrep. The BOLD reference was then co-registered to the T1w reference using bbregister (FreeSurfer), which implements boundary-based registration[32]. Co-registration was

configured with nine degrees of freedom to account for distortions remaining in the BOLD reference. Head-motion parameters with respect to the BOLD reference (transformation matrices, and six corresponding rotation and translation parameters) are estimated before any spatio-temporal filtering using mcflirt (FSL 5.0.9[33]). The BOLD time-series (including slice-timing correction when applied) were resampled onto their original, native space by applying a single, composite transform to correct for head-motion and susceptibility distortions. These resampled BOLD time-series will be referred to as preprocessed BOLD in original space, or just preprocessed BOLD. The BOLD time-series were resampled to MNI152NLin2009cAsym standard space, generating a preprocessed BOLD run in MNI152NLin2009cAsym space. First, a reference volume and its skull-stripped version were generated using a custom methodology of fMRIPrep. Several confounding time-series were calculated based on the preprocessed BOLD: framewise displacement (FD), DVARS and three region-wise global signals. FD and DVARS are calculated for each functional run, both using their implementations in Nipype (following the definitions in ref. 34). The three global signals are extracted within the CSF, the WM, and the whole-brain masks. Additionally, a set of physiological regressors was extracted to allow for component-based noise correction (CompCor[35]). Principal components are estimated after high-pass filtering the preprocessed BOLD time-series for the two CompCor variants: temporal (tCompCor) and anatomical (aCompCor). Six tCompCor components are then calculated from the top 5 percent variable voxels within a mask covering the subcortical regions. This subcortical mask is obtained by heavily eroding the brain mask, which ensures it does not include cortical GM regions. For aCompCor, six components are calculated within the intersection of the aforementioned mask and the union of CSF and WM masks calculated in T1w space, after their projection to the native space of each functional run (using the inverse BOLD-to-T1w transformation). The head-motion estimates calculated in the correction step were also placed within the corresponding confounds file. The BOLD time-series, were resampled to surfaces on the following spaces: fsaverage6. All resamplings can be performed with a single interpolation step by composing all the pertinent transformations (i.e., head-motion transform matrices, susceptibility distortion correction when available, and co-registrations to anatomical and template spaces). Gridded (volumetric) resamplings were performed using antsApplyTransforms (ANTs), configured with Lanczos interpolation to minimize the smoothing effects of other kernels[36]. Many internal operations of fMRIPrep use Nilearn 0.4.2[37], mostly within the functional processing workflow. For more details on the pipeline, see https://fmriprep.org/en/latest/workflows.html.

**Temporal high-pass filter cutoff**. Given that we were looking for an expected slow change in neural pattern (i.e., a change in neural pattern when there is a switch between schemas or extended events), we could not use the default high-pass filter cutoff that is usually used (i.e., 120 s). However, some high-pass filtering was still necessary due to activity that can be attributed to non-neuronal noise (e.g., scanner drift). We preferred to use the longest possible high-pass filter cutoff that still removes the non-neuronal noise. Following the analysis of high-pass filter cutoffs for stimuli with long timescales that was performed in Lositsky et al.[38], we opted for a high-pass filter cutoff of 480 s.

## Analyses
**Statistical methods**. For the representational similarity analysis, we computed one-tailed t values when comparing the observed patterns of neural similarity to predicted patterns. Otherwise, two-tailed *t*-tests were used for all pairwise comparisons except when the Shapiro–Wilk test indicated violations of normality, in which case we used a Wilcoxon signed-rank test. To compute the significance of correlations between neural codes and memory behavior, we used a nonparametric permutation test (see details in "Relating brain activity to behavior" below). We used JASP[39] to compute all *t*-tests, including both frequentist and Bayesian tests. For Bayesian tests, the default prior in JASP was used (Cauchy scale 0.707).

**Behavioral data**. We assessed the learning of the schema in two ways: (1) By calculating a percentage correct for the two-alternative forced choice prediction questions on day 1 as well as day 2. (2) By calculating the percentage that people allocated to the correct answer option during the schema learning test at the end of day 2 and comparing that to the percentage people allocated to the opposite answer option. In addition to checking the schema learning throughout and at the end of the experiment, we assessed whether people were sufficiently attentive throughout the tasks. This was done by calculating a percentage correct for the two-alternative forced choice detail questions at the end of each stimulus during video watching. Lastly, we assessed how well people remembered the episodic details of the weddings. This was done by analyzing the behavioral data acquired during the recall task in the following ways:

- By manually scoring which episodic details were mentioned during recall by each of the participants for each of the weddings separately, leading to a score for each participant and each wedding regarding the amount of correctly and incorrectly remembered details. To get to a score for their memory for details, we then subtracted the number of incorrectly remembered details from the number of correctly remembered details.
- By manually scoring which ritual types were mentioned during recall for each participant and for each of the weddings separately, leading to a score for each participant and each wedding regarding the amount of correctly and the amount of incorrectly mentioned ritual types. To get to a score for their memory for ritual types, we then subtracted the number of incorrectly remembered ritual types from the number of correctly remembered ritual types.

For scoring the amount of details remembered, we compared participants' recalls to a list of possible details that participants could mention (see EpisodicDetailsList_ManualScoring.pdf on https://osf.io/u3cfr/ for this list).

**Representational similarity analysis**. We hypothesized that the brain uses several types of codes to represent the context-dependent temporal structure in this study, including a schema (North/South) code, a path code, and a rotated preceding ritual code (Figs. 2 and 3B), as we will describe in more detail below. To calculate each of these codes, we analyzed the multivoxel pattern of neural activity by applying a searchlight approach on our whole-brain data. Specifically, we used representational similarity analysis (RSA) to examine Pearson's correlation coefficients between patterns of activity within searchlights throughout the whole brain (the searchlights each had a radius of 2 voxels excluding the center voxel—thus, each searchlight encompassed $((2 \times 2) + 1)^3 = 125$ voxels). For these analyses, we used the fMRI data from the part of the wedding videos during which participants watched the actual rituals (i.e., campfire/flower, which below is referred to as stage 2, coin/torch, which below is referred to as stage 3, and egg/painting, which below is referred to as stage 4). We calculated an average neural pattern for each combination of ritual and schema (12 patterns total: [campfire, flower, coin, torch, egg, painting] X [North, South]) across all weddings but one (we refer to these as the 12 "template patterns"), and then compared these 12 template patterns to the average neural patterns for each of the 3 rituals of the held-out wedding (Fig. 3A); this yields 36 (12 × 3) comparisons in total. We computed these 36 comparisons for each possible held-out wedding. For the analyses in the paper, we sorted the comparisons according to the following criteria:

- First, the comparisons were sorted according to the stage of the template ritual (2, 3, 4) and the stage of the held-out ritual (2, 3, 4). For some analyses, we also grouped these more generally based on whether the comparison was *within-stage* (e.g., template = stage 2, held-out ritual = stage 2) or *across stage* (e.g., template = stage 2, held-out ritual = stage 3).

**Fig. 2 | The four different hypothesized codes.**
**A** Schema code, representing North schema vs South schema. **B** Path code, representing each of the 4 sequences separately. **C** Current ritual code, representing each of the six rituals separately. **D** Rotated preceding ritual code, representing a rotated version of the preceding ritual along with the (non-rotated) current ritual. The rotation symbol indicates when we predict a rotated version of the preceding ritual to be present.

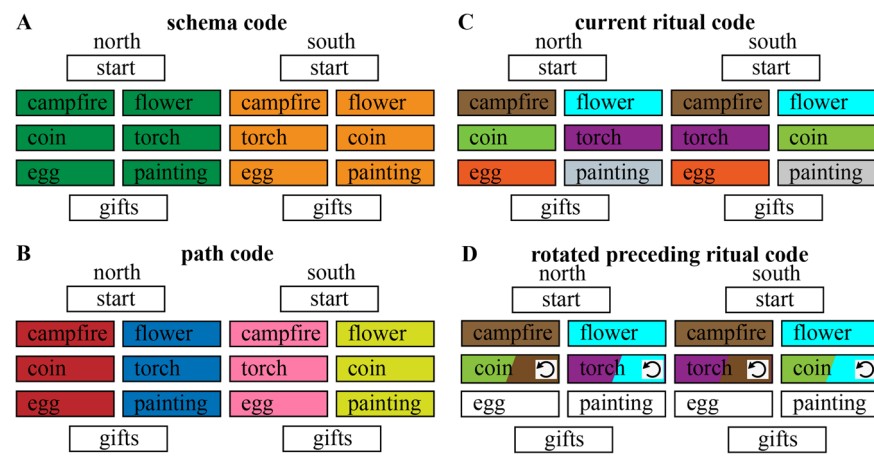

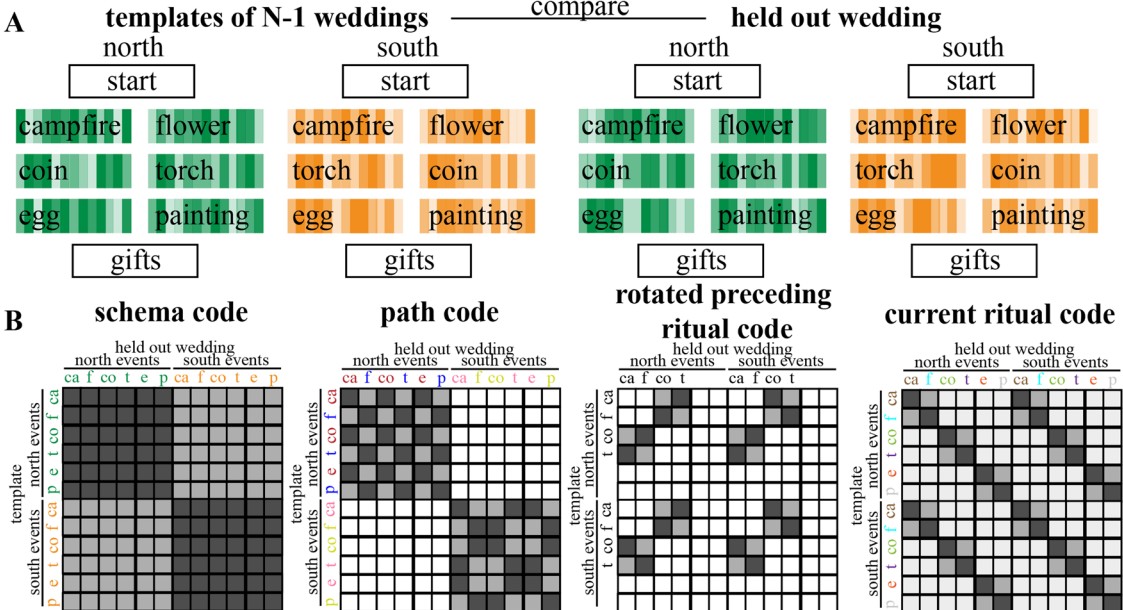

**Fig. 3 | Explanation of the procedure used for the searchlight RSA. A** We calculated an average neural pattern for each combination of ritual (campfire, flower, coin, torch, egg, painting) and schema across all weddings but one (referred to as "template patterns"), and then compared these template patterns to the neural patterns associated with each ritual of a single left-out-wedding; we repeated this process, holding out a different wedding each time. **B** Matrices that show the 4 types of neural codes. Light gray means low similarity is predicted and dark gray means high similarity is predicted. White squares were omitted from the analysis.

- Second, the comparisons were also sorted according to the relationship between the path of the template ritual and the path of the held-out ritual: There are 4 possibilities here: (1) same path and schema; (2) different path, same schema; (3) different schema, none of the rituals on the path of the held-out ritual match the template ritual (e.g., template is north-coin, path of held-out-ritual is south-campfire-torch-egg); (4) different schema, path of held-out ritual matches the template ritual for within-stage comparisons but not across-stage comparisons (e.g., template is north-coin, path of held-out-ritual is south-flower-coin-painting).
  For some analyses, we also noted the following situations:
- whether the template ritual and the held-out ritual had the *same preceding ritual*, even if the template and the held-out ritual were not themselves the same (e.g., torch-North and coin-South are both preceded by flower).
- whether the template ritual matched the ritual that immediately preceded or followed the held-out wedding; e.g., if the template is campfire-North, and the held-out ritual is coin-North, then the template ritual does not match the held-out ritual, but the template

ritual matches the ritual that *immediately preceded* the held-out ritual (this is relevant for the "rotated preceding ritual code" as described below)

Schema code. This code would involve one pattern being active for rituals from North weddings and a distinct pattern being active for rituals from South weddings, regardless of which specific ritual and path is being shown at that time. For the schema code, we separately averaged (within each participant) all comparisons involving the same schema (including both within-stage and across-stage comparisons) and all comparisons involving the other schema (including both within-stage and across-stage comparisons). We then computed, for each participant, the average difference in similarity between same-schema and other-schema comparisons, and assessed the reliability of this difference across participants. Figure 3B shows this contrast in RSA matrix form. We also computed this "schema code" difference score (same-schema minus other-schema) for each individual wedding within each participant (focusing on the regions showing a significant FDR-corrected schema code effect in our RSA analyses), for the analyses described in "Relating brain activity to behavior" below.

Path code. This code would involve distinct patterns being active for rituals from different paths while watching the ceremonies, regardless of which ritual is being shown at that time. For the path code, we computed, for each participant, the difference in similarity between same-path, same-schema and different-path, same-schema comparisons and assessed the reliability of this difference across participants. Figure 3B shows this contrast in RSA matrix form (note that we omitted different-path, different-schema comparisons from this RSA matrix—the benefit of doing this is that the RSA matrix for the path code is uncorrelated with the RSA matrix for the schema code). We also computed this "path code" difference score (same-path, same-schema minus different-path, same-schema) for each individual wedding within each participant (focusing on the regions showing a significant FDR-corrected path code effect in our RSA analyses), for the analyses described in "Relating brain activity to behavior" below.

Rotated preceding ritual code. With this code type, we expect that the stage 3 pattern will contain a rotated representation of the preceding (stage 2) ritual. This implies that similarity values should be *especially low* when the template applied to the stage 3 pattern matches the stage 2 ritual that preceded it. Conversely, similarity values should also be especially low when the template applied to the stage 2 pattern matches the stage 3 ritual that followed it. To compute this, we first looked at cross-stage comparisons where stage 2 templates were matched with held-out stage 3 rituals, and we computed the difference between similarity values when the template applied to the stage 3 pattern *mismatched* the stage 2 ritual that preceded it, vs. when it *matched* the stage 2 ritual that preceded it. Next, we looked at cross-stage comparisons where stage 3 templates were matched with held-out stage 2 rituals, and we computed the difference between similarity values when the template applied to the stage 2 pattern *mismatched* the stage 3 ritual that followed it, vs. when it *matched* the stage 3 ritual that followed it. We then averaged (within participants) these two differences to create a single "rotated preceding ritual code" for each participant, and assessed the reliability of this difference across participants. To get the most clean measure of the rotated preceding ritual code, we excluded stage 4 comparisons (since one would expect residual codes from multiple preceding stages during stage 4, which is not the case yet in stage 2 and 3). Figure 3B shows this contrast in RSA matrix form. We also computed this "rotated preceding ritual code" difference score for each individual wedding within each participant (focusing on the regions showing a significant FDR-corrected rotated preceding ritual code effect in our RSA analyses), for the analyses described in "Relating brain activity to behavior" below.

Current ritual code. In addition to the history-dependent codes described above, we also computed a simple measure of how well the current ritual is being represented. For this "current ritual" code, we separately averaged (within each participant) all within-stage comparisons involving the same ritual and all comparisons involving the other ritual for that stage. We then computed, for each participant, the average difference in similarity between same-ritual and different-ritual comparisons, and assessed the reliability of this difference across participants. Figure 3B shows this contrast in RSA matrix form. We also computed this "current ritual code" difference score (same-ritual minus other-ritual) for each individual wedding within each participant (focusing on the regions showing a significant FDR-corrected current ritual code effect in our RSA analyses), for the analyses described in *Relating brain to behavior* below. Note that the current ritual code, defined this way, is "history-agnostic"—a region that scores well according to this metric may or may not also score well on other, history-dependent codes (e.g., the rotated preceding ritual code described above). In particular, note that the current ritual code only looks at *within-stage* comparisons and the rotated preceding ritual code only looks at *across-stage* comparisons. Since they rely on distinct comparisons, there is no strict mathematical dependence between these codes; in principle, a region could score well according to the current ritual code but not the rotated preceding ritual code, or vice-versa, or it could score well on both, or neither.

Conjunction and disjunction analyses. To determine which regions coded for both the current ritual and the rotated preceding ritual, we overlaid the two brain maps and determined which regions were FDR-corrected significant (with a threshold of $p = 0.05$) for both of these codes. To determine which regions exclusively coded for either the rotated code or the current ritual code, we ran two disjunction analyses. These disjunction analyses determined which regions pass an FDR correction for one of the brain maps (current/rotated preceding) and are above $p = 0.1$ uncorrected for the other brain map.

Significance. To determine statistical significance for the RSA analyses, we used the multipletests function from the statsmodels python package to convert uncorrected $p$-values to $q$-values of the whole brain searchlights of all codes described above (i.e., FDR correction). For visualization of the brain results, we used surfplot[40,41].

**Relating brain activity to behavior**. We related the brain results to individual participants' behavior. The behavioral measures that we used to measure memory for specific weddings were as follows:
- Correct minus incorrect episodic details remembered
- Correct minus incorrect rituals remembered

Incorrect details/rituals here refers to a participant explicitly mentioning a detail/ritual that relates to another wedding—for example, a ritual that was not performed at that wedding or an episodic detail that did not belong to that wedding (for a list, see EpisodicDetailsList_ManualScoring.pdf on https://osf.io/u3cfr/).

We related the brain results to individual participants' behavior. First, for each wedding within each participant, we computed the "neural strength" of each code by running the RSA contrast for that code (see "Representational similarity analysis" section above) on the data from that wedding. We then computed the Spearman correlation (across weddings, within each participant) between each of our neural and behavioral measures. For a given pairing of neural and behavioral measures (e.g., schema code strength vs. memory for ritual types), this yielded one Spearman correlation coefficient per participant, which we then averaged across participants. To test for the statistical significance of these correlations, we calculated a null distribution by shuffling the behavioral scores and re-computing the correlations 1000 times, and then we compared the actual average correlation score to the null distribution of average correlation scores. The $p$-value was computed as the proportion of the null distribution above the actual average correlation score; this is a one-tailed test, instantiating the directional hypothesis that stronger activation of the neural code would lead to better subsequent memory performance. To compute Bayes Factor scores for these correlation analyses, we ran a one-sample, one-tailed Bayesian $t$-test on the Fisher-Z transformed correlation values for each participant (testing the hypothesis that these transformed correlation values are above zero).

## Results

Our experiment relies on participants' ability to learn the schemas that we presented. We used blocked schema training, as Beukers et al.[18] showed that this is effective in the paradigm that we used. To verify that participants learned the schemas, we tested whether participants learned to distinguish the ritual order in the North schema as opposed to the South schema. To assess their learning, participants received stop-and-ask two-alternative forced choice questions asking them to predict what will happen next throughout the task. Additionally, they were tested on their knowledge of the schemas at the end of day 2 using questions like, "If a couple from the North just planted a flower, what is the most likely ritual to happen next?". They were given all possible rituals as answer options and were told to allocate a total of 100 percent across these answer options.

Results from these tasks indicated that participants indeed learned the two schemas (Fig. 4), assigning a significantly higher percentage to the

correct answer option as opposed to the answer option of the opposite schema (W = 807, *p* < 0.001, rb = 0.968, 95% CI = 0.936, 0.984).

## Various neural codes to represent context-dependent sequences

We used a representational similarity analysis (RSA[42]) searchlight approach to discover which brain regions would represent a certain hypothesized code. We calculated an average neural pattern for each of the combinations of ritual + schema (e.g., "campfire + North") across all weddings but one (referred to as "template patterns"), and then compared these templates to the neural patterns evoked by rituals in the left-out-weddings. Then we looped through all weddings (see "Methods" for details).

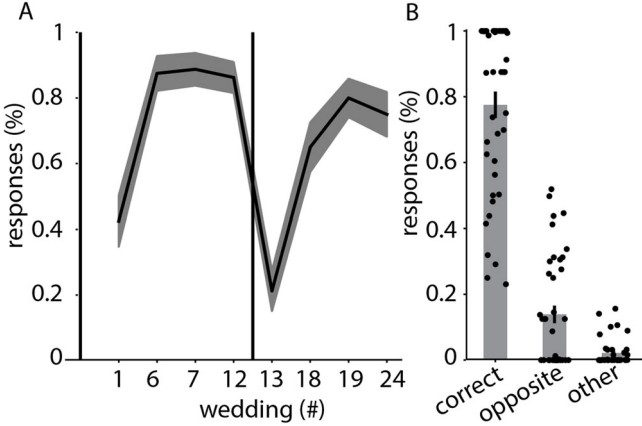

**Fig. 4 | Participants (*N* = 40) learned to distinguish North from South schema. A** Amount of correct answers on 2 alternative forced choice questions throughout the learning task on Day 1. The vertical line between wedding 12 and 13 represents the switch from one to the next schema. **B** percentage assigned to the correct answer option ("correct"), the opposite answer option ("opposite") and the 4 other answer options ("other") during the test for schema learning at the end of Day 2. The percentage for "other" is divided by 4 since it contains 4 out of 6 answer possibilities. Error ribbons (in **A**) and bars (in **B**) indicate the standard error of the mean.

**Schema code across the brain.** For the schema code, we expected that different neural patterns would be active for rituals that belong to North or South weddings, regardless of which specific ritual and path was being shown at that time. This was calculated by contrasting the similarity of neural patterns corresponding to the same schema and neural patterns corresponding to different schemas (see Fig. 3 and "Methods"). The whole-brain RSA searchlight approach showed a significant schema code (FDR corrected) in several regions, most prominently in the thalamus, pallidum, caudate, posterior medial cortex, hippocampus, para-hippocampus, fusiform gyrus, inferior and superior temporal regions, temporal pole, superior frontal regions (Fig. 5A). Figure 6, Supplementary Fig. 1, and Supplementary Table 1 show a post hoc confirmation of the pattern.

**Path code in postcentral gyrus.** For the path code, we expected that different neural patterns would be active for each of the four different sequences ("paths"). This was calculated by contrasting the similarity of neural patterns belonging to the same path and neural patterns belonging to the other path from the same schema (see Fig. 3 and "Methods"). The whole-brain RSA searchlight approach showed a significant path code (FDR-corrected) in postcentral gyrus (Fig. 5B). Figure 7, Supplementary Fig. 2, and Supplementary Table 2 show a post hoc confirmation of the pattern.

**Rotated preceding ritual code and current ritual code in medial occipital regions.** For this code, we expected: (1) that the preceding ritual would be represented as a rotated (photo-negative) version of how that ritual is represented when it is currently unfolding, and (2) that the current ritual would be represented. To identify regions with the rotated preceding code property, we compared templates acquired at the campfire/flower stage to patterns acquired at the coin/torch stage (and vice-versa). Specifically, we looked for regions where patterns during the coin/torch stage were *especially dissimilar* to the template corresponding to the preceding ritual (e.g., if coin were preceded by campfire, we would expect it to be *less similar* to the campfire template than the flower template)—the prediction of there being *less* rather than *more* similarity is due to the hypothesized "photo-negative" nature of the code (see Fig. 3

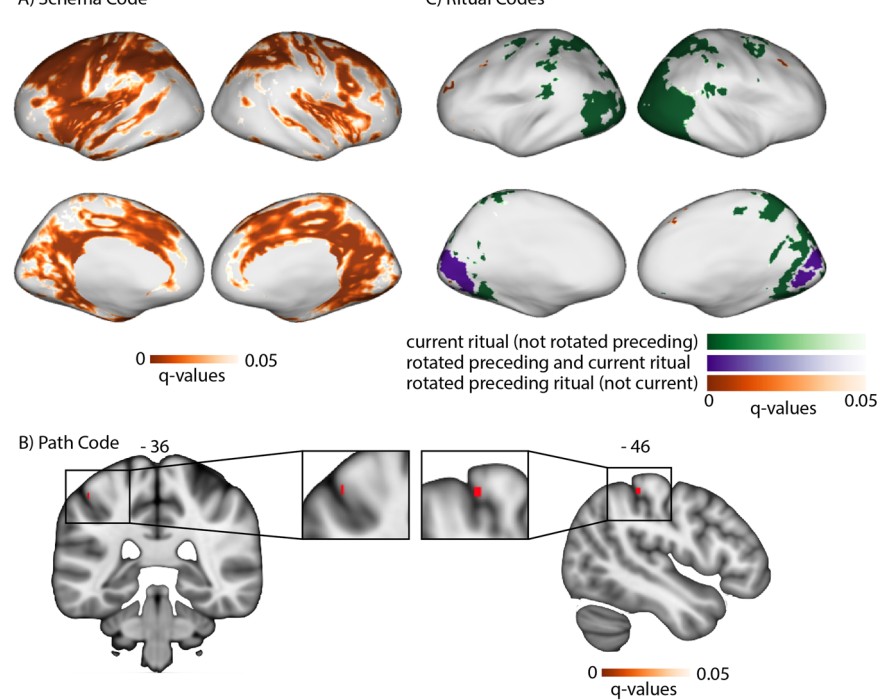

**Fig. 5 | Results of the searchlight RSA showing FDR-corrected brain maps for the 5 neural codes (*N* = 40). A** FDR-corrected brain map for the schema code. **B** FDR-corrected brain map for the path code. **C** FDR-corrected brain maps for both rotated preceding and current ritual (blue), current ritual exclusively (green) and rotated ritual exclusively (red). In more detail: (blue) FDR-corrected brain map for those regions that pass an FDR correction for rotated preceding ritual code and for current ritual code. (red) FDR-corrected brain map for rotated preceding ritual code only (i.e., removing brain regions that also show a current ritual code at liberal threshold of *p* < 0.1 uncorrected). (green) FDR-corrected brain map for current ritual code only (i.e., removing regions that also show a rotated preceding ritual code at liberal threshold of *p* < 0.1 uncorrected).

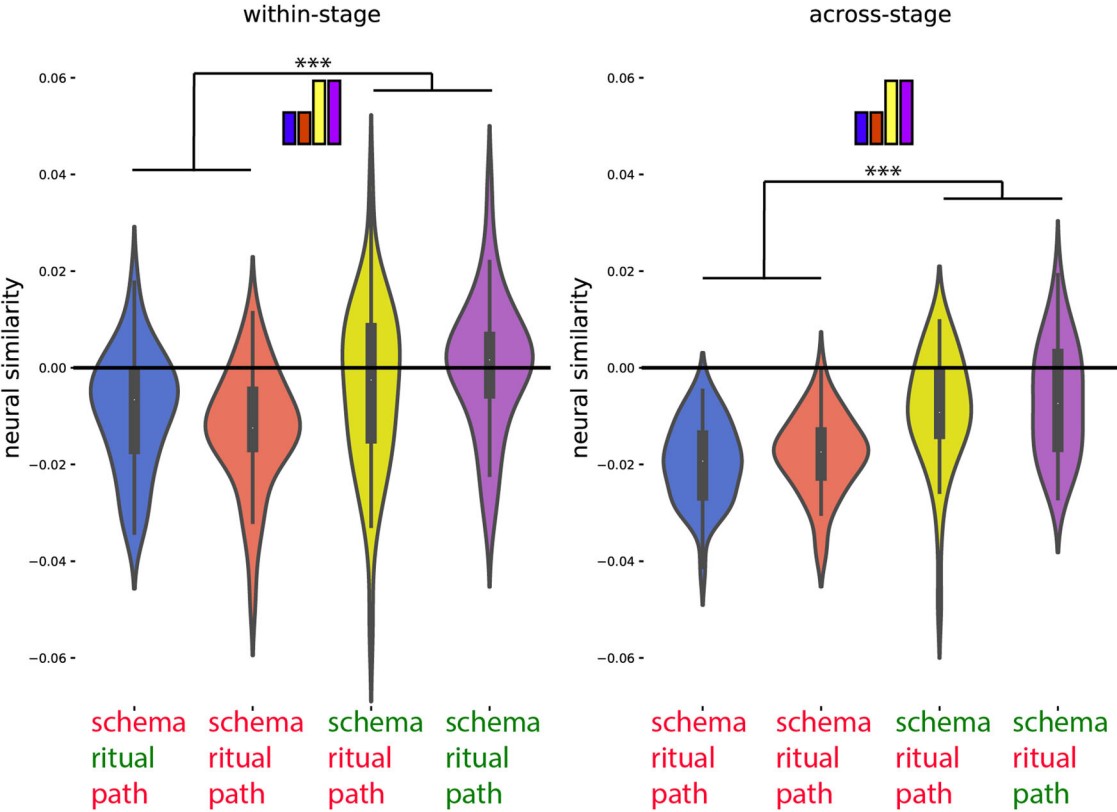

**Fig. 6 | Post hoc confirmation of the pattern for the regions identified in Fig. 5 as showing the schema neural code ($N = 40$).** Colors of violin plots indicate the relationship between the paths of the template and the held-out ritual: Purple = same path; yellow = different path, same schema; orange = different schema, always different ritual; blue = different schema, same ritual for within-stage comparisons (but not across-stage comparisons). For convenience, colors of x-axis labels indicate whether the schema/ritual/path are the same (green) or different (red) for the template and held-out ritual. [left] An average of all within-stage comparisons. [right] An average of all across-stage comparisons. The small bars at the top of each plot indicate the predicted pattern. The schema code pattern was present for both within-stage and across-stage comparisons, as evidenced by higher similarity for same-schema conditions (i.e., yellow and purple) vs. different-schema conditions

(i.e., blue and orange); within-stage: $t(39) = 4.214$, $p < 0.001$, $d = 0.666$, 95% CI = 0.320, 1.006; across-stage: $t(39) = 6.310$, $p < 0.001$, $d = 0.998$, 95% CI = 0.613, 1.374. While these regions showed higher similarity for same vs. different schemas, they did not show significant sensitivity to same vs. different path: Paired-samples t-tests comparing same-path, same-schema (purple) to different-path, same-schema (yellow) were nonsignificant both within-stage and across-stage, and BF-01 values (Bayes Factor scores in favor of the null hypothesis) indicated moderate evidence for the null for the across-stage analysis; within-stage: $t(39) = 1.230$, $p = 0.226$, $d = 0.195$, 95% CI = −0.120, 0.506, BF-01 = 2.911, across-stage: $t(39) = 0.690$, $p = 0.494$, $d = 109$, 95% CI = −0.202, 0.419, BF-01 = 4.690. See Supplementary Fig. 1 for a more detailed version of the result. See Supplementary Table 1 for full statistics. Black asterisks in the figure indicate significance: \*\*\*$p < 0.001$.

and "Methods"). To identify regions with the current ritual property, we contrasted (within a particular stage of the wedding) the similarity of neural patterns corresponding to the same ritual (regardless of schema) and neural patterns corresponding to different rituals (regardless of schema; see Fig. 3 and "Methods"). Next, we overlaid these two brain maps and determined which brain regions are FDR-corrected significant for both of these properties. This approach showed a significant rotated preceding ritual code (FDR-corrected) and current ritual code (FDR-corrected) most prominently in medial occipital regions (Fig. 5C, blue color map). Figure 8 and Supplementary Table 3 show a post hoc confirmation of the pattern.

Importantly, the pattern of results that we attribute above to a rotated *preceding* ritual code can be explained equally well by a rotated *upcoming* ritual code: When the template is made based on the stage 2 ritual and then applied to the stage 3 ritual (or vice-versa), both the rotated prospective (upcoming-ritual) and retrospective (preceding-ritual) accounts predict that pattern similarity should be reduced when the template is the successor or predecessor of the held-out ritual (for the prospective code, the preceding ritual should contain a "photo-negative" of the following ritual; for the retrospective code, the following ritual should contain a "photo-negative" of the preceding ritual; either way, similarity should be reduced). However, the expected pattern of RSA results when the template is made based on stage 2 and then applied to stage 2 will differ depending on whether the code is

prospective or retrospective. If the code is prospective, then rituals with the same successor should show an elevated correlation. As shown in Fig. 8 and Supplementary Table 3, this pattern of results was not present, with the Bayes Factor (BF-01) score = 3.438, indicating moderate support for the null. This aligns with our original hypothesis that this region is coding for rotated preceding rituals, not upcoming rituals.

**Regions that code exclusively for rotated preceding or current ritual.** As described above, medial occipital regions coded for both the rotated preceding ritual and for the current ritual. To identify regions that coded for the rotated preceding ritual exclusively (i.e., not for current ritual) and regions that coded for the current ritual exclusively (i.e., not for preceding rotated code), we ran two disjunction analyses where a region was included if it passed FDR correction for one code but failed to show the other code at a liberal threshold of $p < 0.1$ uncorrected.

**Rotated preceding ritual code (without current ritual code) in superior and mid frontal regions.** In the first disjunction analysis, we determined which regions pass FDR correction for the rotated preceding ritual code, but fail to show the current ritual code at a liberal threshold of $p < 0.1$ uncorrected. The whole-brain RSA searchlight approach showed a significant rotated preceding but not current ritual code in superior and mid frontal regions (Fig. 5C, red color map). Figure 9 and Supplementary

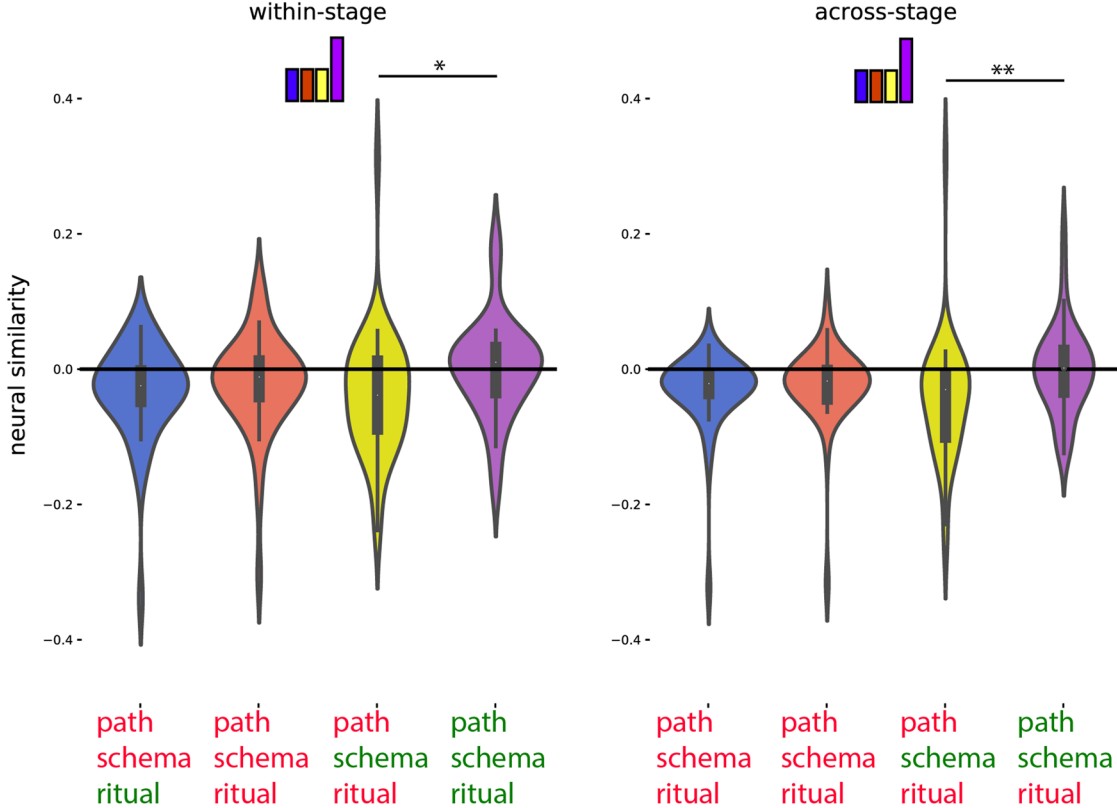

**Fig. 7 | Post hoc confirmation of the pattern for the regions identified in Fig. 5 as showing the path neural code (***N* = 40**).** Colors of violin plots indicate the relationship between the paths of the template and the held-out ritual: Purple = same path; yellow = different path, same schema; orange = different schema, always different ritual; blue = different schema, same ritual for within-stage comparisons (but not across-stage comparisons). For convenience, colors of *x*-axis labels indicate whether the schema/ritual/path are the same (green) or different (red) for the template and held-out ritual. [left] An average of all within-stage comparisons. [right] An average of all across-stage comparisons. The small bars at the top of each plot indicate the predicted pattern. These regions show a path code pattern for both within-stage and across-stage comparisons, as evidenced by higher similarity for the

same-path, same-schema condition (i.e., purple) compared to the different-path, same-schema condition (i.e., yellow); within-stage: t(39) = 2.514, *p* = 0.016, *d* = 0.397, 95% CI = 0.073, 0.717; across-stage: t(39) = 3.109, *p* = 0.003, *d* = 0.492, 95% CI = 0.160, 0.817. The presence of a significant across-stage path code effect is especially important—the within-stage path code effect is confounded with same vs. different ritual so it does not provide strong evidence for path coding per se. All of the across-stage comparisons looking at same vs. different path involve different rituals, so they are not subject to this confound. See Supplementary Fig. 2 for a more detailed version of the result. See Supplementary Table 2 for full statistics. Black asterisks in the figure indicate significance: **p* < 0.05; ***p* < 0.01.

Table 4 show a post hoc confirmation of the pattern. Post hoc, we discovered that these particular regions also appeared to show a schema code (see Fig. 9). The reason for some of these regions not being present in the schema code brain map (i.e., Fig. 5A) is that Fig. 5A is corrected for whole-brain multiple comparison correction, whereas the schema pattern in Fig. 9 is a post hoc test, not corrected for multiple comparisons.

To assess whether this region codes for rotated preceding rituals (as hypothesized) or upcoming rituals, we looked at the RSA results when the template is made based on stage 2 and then applied to stage 2. As described above, if the region is coding for upcoming rituals, then rituals with the same successor should show an elevated correlation. As shown in Fig. 9 and Supplementary Table 4, the correlation was actually lower for rituals with the same successor, which is incompatible with this region coding for the upcoming ritual.

**Current ritual code (history-agnostic, without rotated preceding ritual code) in lateral occipital regions.** In the second disjunction analysis, we looked for regions that represent the current ritual while participants are watching the weddings (FDR-corrected), but fail to show the rotated preceding ritual code at a liberal threshold of *p* < 0.1 uncorrected. The whole-brain RSA searchlight approach showed a significant current ritual but not a rotated preceding code in lateral occipital cortex regions (Fig. 5C, green color map). Figure 10, Supplementary Fig. 3, and Supplementary Table 5 show a post hoc confirmation of the pattern.

**Other codes that we searched for.** For completeness, we also searched for a *non-rotated* preceding ritual code; this would be manifest in the representational similarity analysis as the exact opposite of the pattern of similarity scores that indicates a rotated preceding ritual code. As such, we could use the exact same neural contrast, here looking for a large *negative* correlation between the predicted and observed pattern of similarity scores—note that this same contrast would detect both a non-rotated *preceding* ritual code and a non-rotated *upcoming* ritual code (both accounts predict that pattern similarity should be increased when the template is the successor or predecessor of the held-out ritual). No searchlights passed FDR correction for this "non-rotated preceding/upcoming ritual code" analysis. To estimate the level of support for the null hypothesis (i.e., that this code is absent) we also computed log Bayes Factors across the whole brain for this code (see Supplementary Fig. 4).

**Relation between schema code and memory**
Together, these results suggest that different types of history-dependent (and history-agnostic) neural codes are present in the distinct brain regions while viewing temporally extended events. Next, we sought to discover how these neural codes relate to subsequent episodic memory for the weddings. Memory for episodic details was tested with a separate cued recall test that was administered after all of the weddings were shown. In this test, participants were shown a picture of a couple and asked to recall as many details as possible about that couple's wedding. Scoring of participants' memory

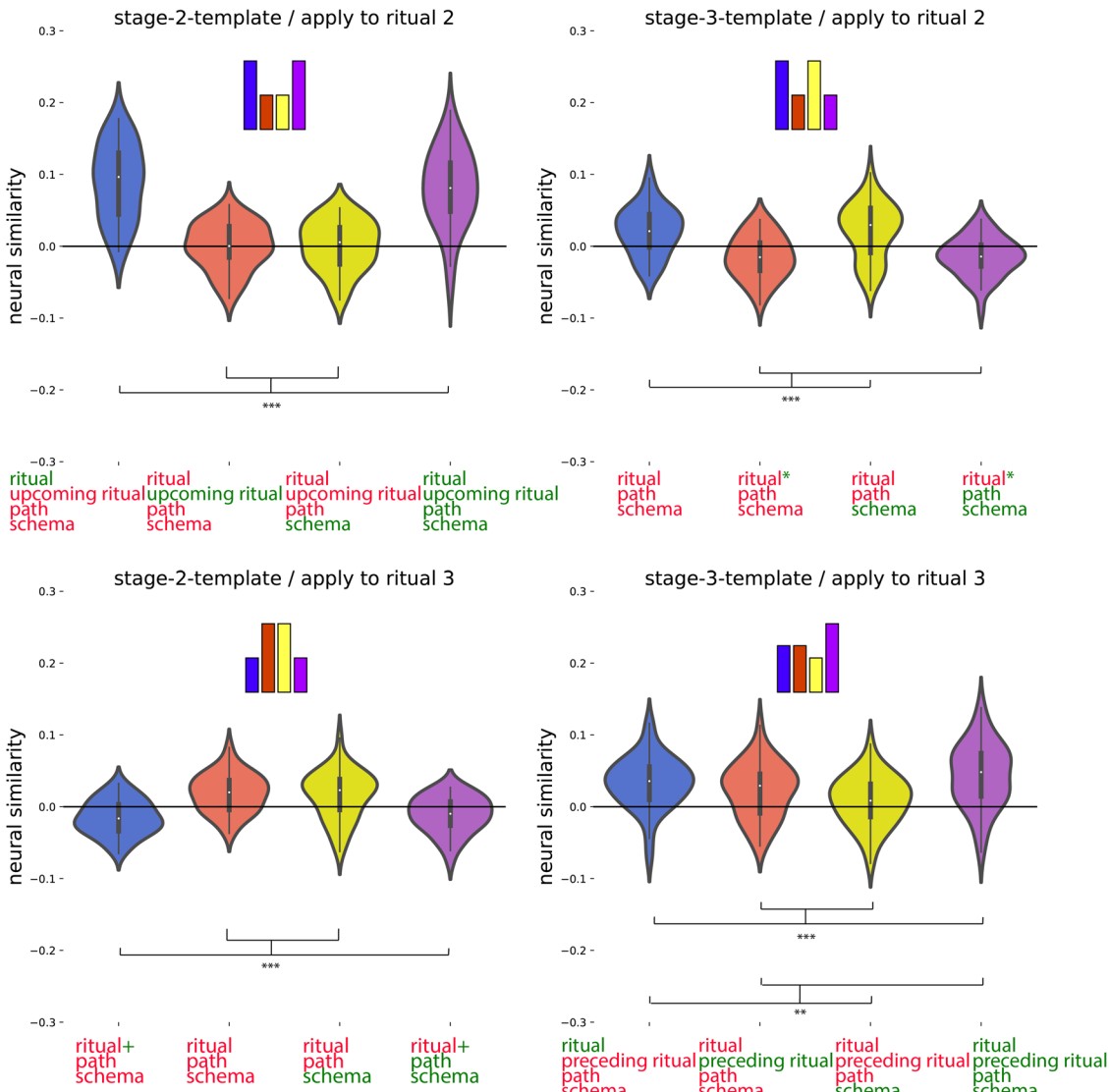

**Fig. 8 | Post hoc confirmation of the pattern for the regions identified in Fig. 5 as showing both the rotated preceding ritual and the current ritual neural code (N = 40).** Colors of violin plots indicate the relationship between the paths of the template and the held-out ritual: Purple = same path; yellow = different path, same schema; orange = different schema, always different ritual; blue = different schema, same ritual for within-stage comparisons (but not across-stage comparisons). For convenience, colors of x-axis labels indicate whether the schema/ritual/preceding-ritual/path are the same (green) or different (red) for the template and held-out ritual. * in green indicates that the held-out pattern corresponds to the ritual that *immediately preceded* the template ritual. + in green indicates that the held-out pattern corresponds to the ritual that *immediately followed* the template ritual. The 4 sub-figures correspond to different combinations of using stage 2/3 as the template and applying the template to stage 2/3. The small bars at the top of each plot indicate the predicted pattern. These regions indeed show a rotated code pattern, as evidenced by the following tests across the four subplots: First, in the upper-right plot, there was lower similarity for orange and purple (held-out ritual immediately preceded the template ritual) than blue and yellow, t(39) = 5.770; p < 0.001, d = 0.912, 95% CI = 0.538, 1.278. In the lower-left plot, there was lower similarity for blue and

purple (held-out ritual immediately followed the template ritual) than orange and yellow, t(39) = 5.818; p < 0.001, d = 0.920, 95% CI = 0.545, 1.286. The upper-left plot provides evidence that these regions also coded for the current ritual: Similarity was higher for same-ritual (i.e., blue and purple) than across-ritual (i.e., orange and yellow) comparisons, t(39) = 9.383, p < 0.001, d = 1.484, 95% CI = 1.028, 1.930. There was no significant evidence for a (rotated or non-rotated) upcoming ritual code in the upper-left plot: The paired-samples t-test comparing conditions with the same upcoming ritual (i.e., purple and orange) to conditions with different upcoming rituals (i.e., yellow and blue) was nonsignificant, and the computed BF-01 value indicated moderate evidence in favor of the null, t(39) = −1.072, p = 0.290, d = −0.169, 95% CI = −0.481, 0.144, BF-01 = 3.438. The lower-right plot provides additional evidence that these regions coded for both the current ritual and the preceding ritual: Similarity was higher when the current ritual was the same (purple and blue) than it was different (yellow and orange), t(39) = 4.204, p < 0.001, d = 0.665, 95% CI = 0.318, 1.004. Also, similarity was higher when the preceding ritual was the same (purple and orange) than when it was different (yellow and blue), t(39) = 2.698, p = 0.01, d = 0.427, 95% CI = 0.100, 0.748. See Supplementary Table 3 for full statistics. Black asterisks in the figure indicate significance: **p ≤ 0.01; ***p < 0.001.

performance was done manually; we separately scored memory for unique episodic details and for the ritual types that comprised the wedding (see "Methods" for details). To relate these memory scores to neural data, we first computed the strengths of each of the neural codes (schema, path, rotated preceding ritual, current ritual) on a per-wedding basis within participants. Then we correlated each of these neural scores with each of the behavioral measures (across weddings, within participants, using Spearman

correlation) and averaged the correlations across participants. We computed significance using a permutation test of the (one-tailed) hypothesis that the strength of the neural code was positively related to memory behavior (see "Methods" for details).

Our results showed that the strength of the neural schema code was significantly correlated with both of our measures of subsequent episodic memory (Fig. 11, Table 1). None of the other correlations between neural

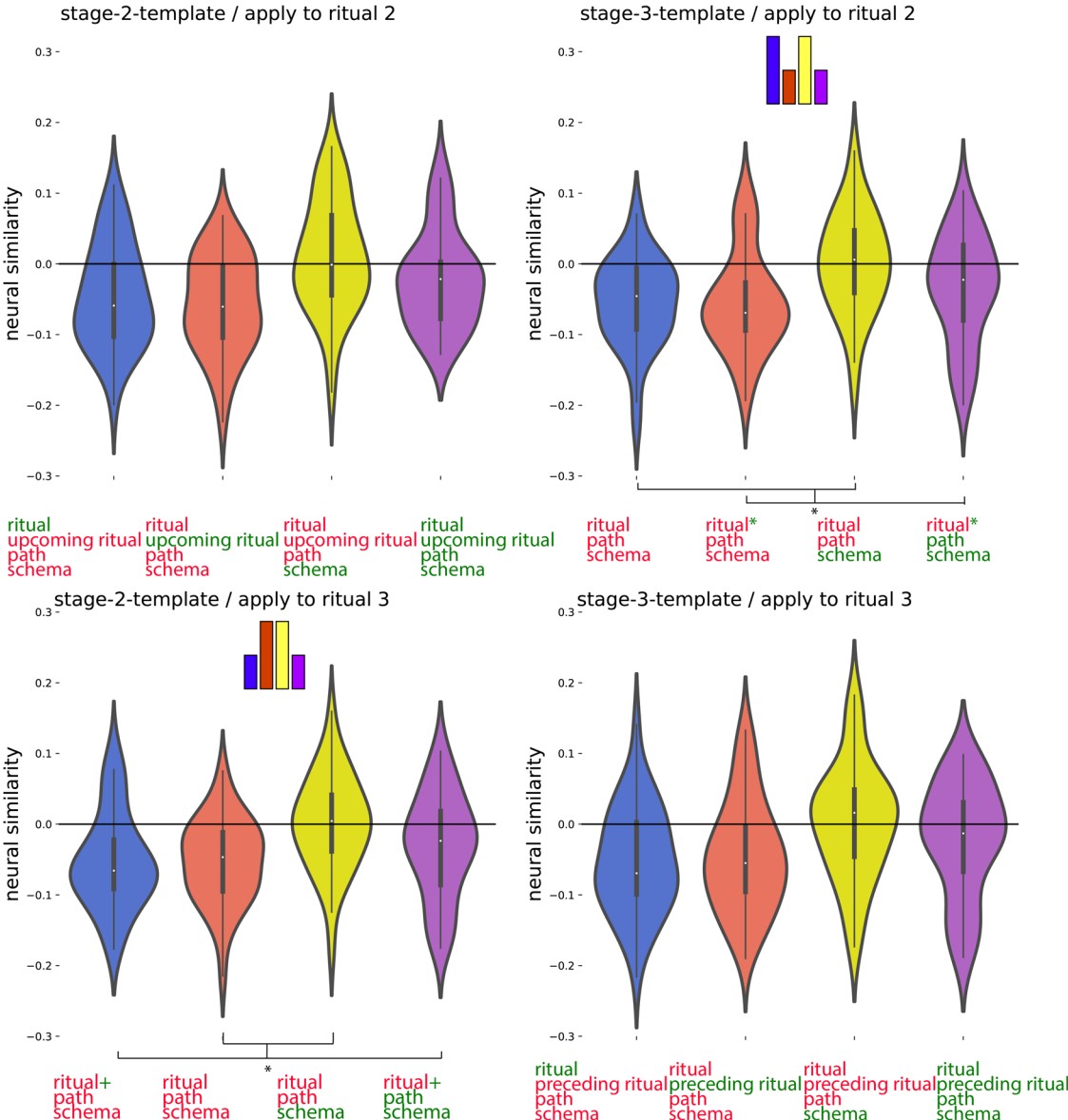

**Fig. 9 | Post hoc confirmation of the pattern for the regions identified in Fig. 5 as showing the rotated preceding ritual neural code but not the current ritual neural code (N = 40).** Colors of violin plots indicate the relationship between the paths of the template and the held-out ritual: Purple = same path; yellow = different path, same schema; orange = different schema, always different ritual; blue = different schema, same ritual for within-stage comparisons (but not across-stage comparisons). For convenience, colors of x-axis labels indicate whether the schema/ritual/preceding-ritual/path are the same (green) or different (red) for the template and held-out ritual. * in green indicates that the held-out pattern corresponds to the ritual that *immediately preceded* the template ritual. + in green indicates that the held-out pattern corresponds to the ritual that *immediately followed* the template ritual. The 4 sub-figures correspond to different combinations of using stage 2/3 as the template and applying the template to stage 2/3. The small bars at the top of each plot indicate the predicted pattern. These regions indeed show a rotated preceding ritual code, as evidenced by the following tests across the subplots: First, in the

upper-right plot, there was lower similarity for orange and purple (held-out ritual immediately preceded the template ritual) than blue and yellow, W = 254; $p$ = 0.036, rb = −0.380, 95% CI = −0.639, −0.045. In the lower-left plot, there was lower similarity for blue and purple (held-out ritual immediately followed the template ritual) than orange and yellow, W = 264; $p$ = 0.05, rb = −0.356, 95% CI = −0.622, −0.017. To test for a (rotated or non-rotated) upcoming ritual code in the upper-left plot, we compared conditions with the same upcoming ritual (i.e., purple and orange) to conditions with different upcoming rituals (i.e., yellow and blue). The comparison was actually significant in the opposite direction, which is incompatible with the presence of an upcoming ritual code; W = 621, $p$ = 0.004, rb = 0.515, 95% CI = 0.210, 0.728. Post hoc, these regions also showed a schema code pattern, as evidenced by high similarity for same-schema (yellow and purple) versus different-schema (blue and orange) comparisons when averaging across the four subplots, W = 568; $p$ = 0.033, rb = 0.385, 95% CI = 0.051, 0.642. See Supplementary Table 4 for full statistics. Black asterisks in the figure indicate significance: *$p$ ≤ 0.05.

codes and behavioral measures were significant. To enable a better interpretation of these nonsignificant results, we also computed Bayes Factor scores that quantify the evidence in favor of the null hypothesis (BF-01); for all of the nonsignificant results, BF-01 scores indicated moderate or strong support for the null hypothesis (i.e., that activation of the neural code is not associated with improved recall; see Table 1 for BF-01 values). We also assessed, post hoc, whether the correlations between the neural schema code

and behavior were *significantly larger* than the correlations between other neural codes and behavior. Only one of these comparisons was significant as measured using a two-tailed paired-samples *t*-test (the correlation with memory for rituals was larger for the schema code than for the rotated preceding ritual code, measured from regions that do not also code for the current ritual). The rest of the comparisons were not significant; BF-01 scores are reported below. For memory for details: schema > path, t(39) =

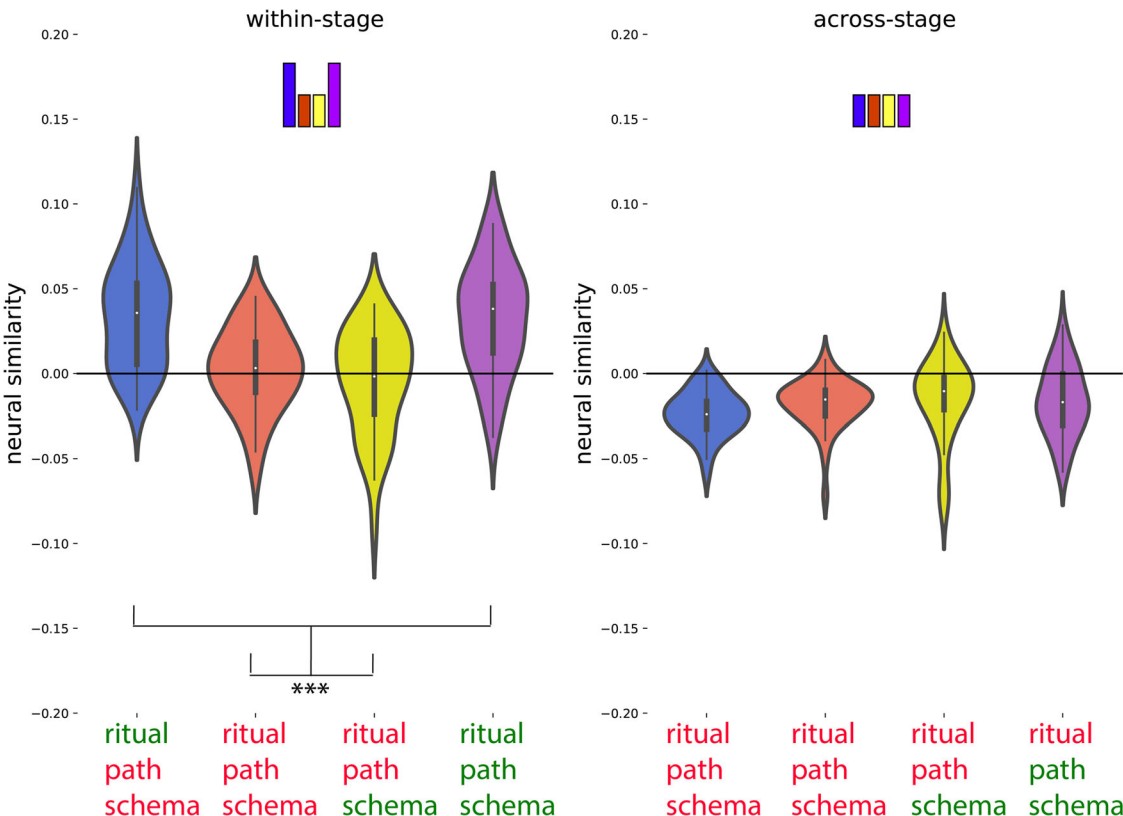

**Fig. 10 | Post hoc confirmation of the pattern of the regions identified in Fig. 5 as showing the current ritual neural code but not the rotated preceding ritual neural code ($N = 40$).** Colors of violin plots indicate the relationship between the paths of the template and the held-out ritual: Purple = same path; yellow = different path, same schema; orange = different schema, always different ritual; blue = different schema, same ritual for within-stage comparisons (but not across-stage comparisons). For convenience, colors of *x*-axis labels indicate whether the schema/ritual/path are the same (green) or different (red) for the template and held-out ritual. [left] An average of all within-stage comparisons. [right] An average of all across-stage comparisons. The small bars at the top of each little plot indicate the predicted pattern. The current ritual code pattern was present for within-stage comparisons, as evidenced by higher similarity for same-ritual conditions (i.e., blue and purple)

compared to different-ritual conditions (i.e., yellow and orange); t(39) = 8.796, $p < 0.001$, $d = 1.391$, 95% CI = 0.950, 1.823. We did not expect there to be differences between these conditions for across-stage comparisons, because the current ritual is always different in across-stage comparisons. In line with this prediction, there were no significant differences between blue and purple when compared to yellow and orange, and the BF-01 score indicated moderate support for the null hypothesis, t(39) = −0.956, $p = 0.345$, $d = -0.151$, 95% CI = −0.462, 0.161, BF-01 = 3.829. See Supplementary Fig. 3 for a more detailed version of the result. See Supplementary Table 5 for full statistics. Black asterisks in the figure indicate significance: ***$p < 0.001$.

1.335, $p = 0.19$, $d = 0.211$, 95% CI = −0.104, 0.523, BF-01 = 2.577; schema > rotated preceding ritual (measured from regions that code for both the rotated preceding ritual and the current ritual), t(39) = 1.024, $p = 0.312$, $d = 0.162$, 95% CI = −0.151, 0.473, BF-01 = 3.598; schema > rotated preceding ritual (measured from regions that do not also code for the current ritual), t(39) = 1.455, $p = 0.154$, $d = 0.230$, 95% CI = −0.085, 0.543, BF-01 = 2.217; schema > current ritual (measured from regions that code for the rotated preceding ritual and the current ritual), t(39) = 1.944, $p = 0.059$, $d = 0.307$, 95% CI = −0.012, 0.623, BF-01 = 1.069; schema > current ritual (measured from regions that do not also code for the rotated preceding ritual), t(39) = 1.569, $p = 0.125$, $d = 0.248$, 95% CI = −0.068, 0.561, BF-01 = 1.901. For memory for rituals: schema > path, t(39) = 1.501, $p = 0.142$, $d = 0.237$, 95% CI = −0.078, 0.550, BF-01 = 2.086; schema > rotated preceding ritual (measured from regions that code for both the rotated preceding ritual and the current ritual), t(39) = 1.502, $p = 0.141$, $d = 0.237$, 95% CI = −0.078, 0.550, BF-01 = 2.085; schema > rotated preceding ritual (measured from regions that do not also code for the current ritual), t(39) = 2.746, $p = 0.009$, $d = 0.434$, 95% CI = 0.107, 0.756, BF-01 = 0.226; schema > current ritual (measured from regions that code for the rotated preceding ritual and the current ritual), t(39) = 1.797, $p = 0.08$, $d = 0.284$, 95% CI = −0.034, 0.599, BF-01 = 1.358; schema > current ritual (measured from regions that do not also code for the rotated preceding ritual), t(39) = 1.710, $p = 0.095$, $d = 0.270$, 95% CI = −0.047, 0.584, BF-01 = 1.550.

## Discussion

In this study, we investigated the neural representations that participants use to disambiguate overlapping sequences during narrative perception, and how these representations relate to memory for episodic details. The results showed that the brain represents such context-dependent temporal structures in multiple distinct ways. First, an abstract schema code (representing whether the couple was from the "North" or "South" culture) was present in a network of regions: thalamus, pallidum, caudate, posterior medial cortex, hippocampus, parahippocampus, fusiform gyrus, inferior and superior temporal regions, temporal pole and superior frontal regions. At the same time, postcentral gyrus showed a representation of the specific sequence (path) of the wedding, medial occipital regions represented the preceding ritual in addition to the current ritual, lateral occipital regions represented exclusively the current ritual, and superior/mid frontal regions represented exclusively the preceding ritual. This representation of the preceding ritual in the medial occipital cortex and in superior/mid frontal regions was rotated[4–6], such that the representation of a particular ritual when it occurred in the past was anticorrelated with the representation of that ritual when it was being perceived. Of these four types of neural codes present in the brain while viewing temporally extended events, only the schema code positively correlated (within participants) with participants' memory performance. This positive within-participant correlation was present for memory for ritual types as well as memory for event details. This finding is consistent

with prior work by Masis-Obando, Norman, and Baldassano[15] showing a relation between (1) the strength of schema representations during perception of (audio-only and audiovisual) narratives and (2) subsequent recall of details from those narratives. Altogether, these findings shed light onto the multiple ways that the brain codes for complex and temporally extended events, how it provides structure for such complex events in mind, and how that structure helps to scaffold memory for specific episodic details.

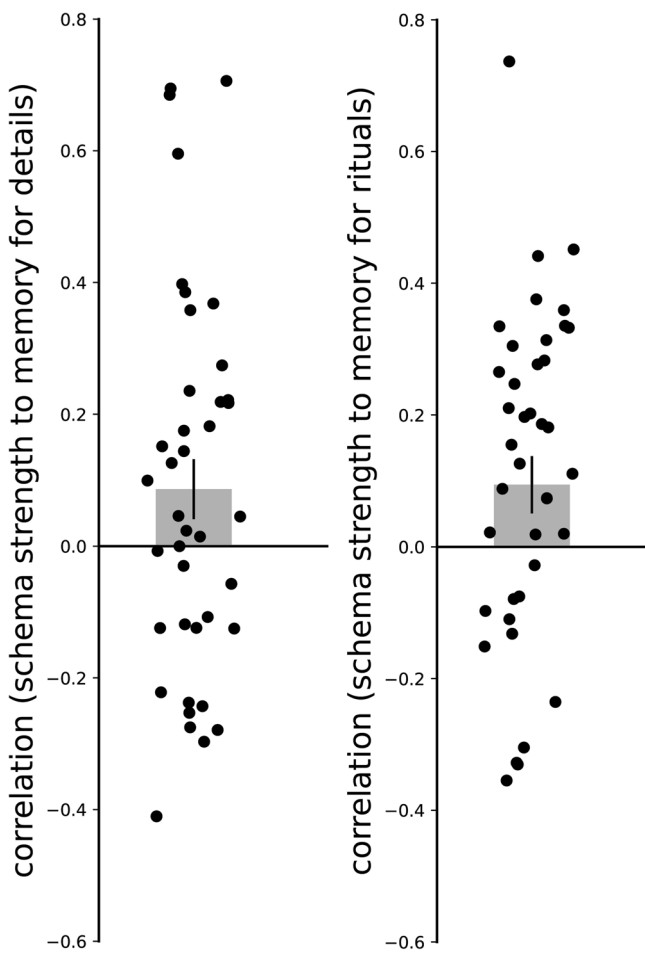

**Fig. 11 | Within-subject correlations (N = 40) of schema strength to memory for details and to memory for rituals.** Error bars indicate the standard error of the mean.

### Relationship to other studies

**Schema code.** As noted above, we found a schema code in a wide range of regions. The most comparable study to ours is Baldassano, Hasson, and Norman[19], which also looked at schema representations for temporally extended narratives. However, the nature of schemas they (and other schema/narrative studies[43]) investigated was different for several reasons. First, we used a novel schema (devised by us), whereas Baldassano, Hasson, and Norman[19] examined schema representations for commonly experienced events like going through airport security and ordering food at a restaurant (see also ref. 15, which reported analyses of the memory retrieval data from this study). Second, these earlier studies[19,43] both have content differences across the schemas, while in our design each ritual is equally often part of each schema. This makes it possible to investigate schema differences without possible confounds due to content differences. These differences between the task designs notwithstanding, it is still useful to compare our results to theirs.

Several of the schema code regions that we observed here were also found by Baldassano, Hasson, and Norman[19], including posterior medial cortex (see also refs. 44–47), superior frontal gyrus (see also refs. 47,48), parahippocampal cortex (see also refs. 20,47,49–52), angular gyrus (see also refs. 20,47,49,52), hippocampus (see also refs. 43,45,50,51,53–56), entorhinal cortex, left subcentral gyrus, postcentral sulcus, left postcentral gyrus, superior temporal sulcus, left posterior temporal sulcus, insula, and prostriata.

While many other studies have found a role for mPFC in schema representation[19,20,46–48,50,51,53,55–64], mPFC was not among the regions that passed FDR correction for the schema code in our study. In the "Limitations" section below, we discuss aspects of our design that may have affected the strength of schema representation in mPFC.

Unlike the Baldassano, Hasson, and Norman[19] study, which only investigated schema codes on the cortical surface and in the hippocampus, we also investigated other subcortical regions. We included the subcortical regions because prior research has established the importance of some subcortical regions (i.e., basal ganglia and thalamus) in supporting interactions between hippocampus and frontal cortex[55]. We indeed found a robust schema code in the thalamus and basal ganglia regions (pallidum, caudate). This is in line with earlier research[43] that found storyline-specific effects in these subcortical areas.

Surprisingly, as outlined above, we also found the schema neural code in the insula. We do not at present have a good account of what role the insula is playing here, but we do note that other studies have found that the insula is involved in representing event schemas[19] and in anticipatory coding of upcoming events[65].

### Table 1 | Within-subject brain-behavior correlations

| Type of code | | r mean ± S.E.M. | p-value | 95% CI | BF-01 |
|---|---|---|---|---|---|
| Schema code | Details | 0.086 ± 0.045 | 0.028* | −0.005, 0.178 | |
| | Rituals | 0.094 ± 0.043 | 0.022* | 0.007, 0.182 | |
| Path code | Details | −0.006 ± 0.049 | 0.56 | −0.105, 0.094 | 6.414 |
| | Rituals | −0.016 ± 0.053 | 0.61 | −0.123, 0.092 | 7.228 |
| Rotated preceding ritual code (measured from regions that show both the current ritual and the rotated preceding ritual code) | Details | 0.002 ± 0.06 | 0.486 | −0.121, 0.124 | 5.747 |
| | Rituals | −0.024 ± 0.057 | 0.654 | −0.140, 0.092 | 7.890 |
| Rotated preceding ritual code (measured from regions that do not also code for the current ritual) | Details | −0.006 ± 0.047 | 0.548 | −0.100, 0.089 | 6.409 |
| | Rituals | −0.089 ± 0.060 | 0.906 | −0.210, 0.033 | 13.413 |
| Current ritual code (measured from regions that show both the current ritual and the rotated preceding ritual code) | Details | −0.018 ± 0.041 | 0.657 | −0.101, 0.065 | 7.951 |
| | Rituals | −0.006 ± 0.048 | 0.561 | −0.103, 0.091 | 6.427 |
| Current ritual code (measured from regions that do not also code for the rotated preceding ritual) | Details | −0.029 ± 0.050 | 0.712 | −0.130, 0.073 | 8.633 |
| | Rituals | −0.026 ± 0.055 | 0.671 | −0.138, 0.086 | 8.095 |

**Rotated preceding ritual code**. Besides a schema code, we also discovered a rotated code in which the sensory representation (current ritual) is anticorrelated to the memory representation (preceding ritual). Interestingly, medial occipital regions showed this rotated preceding ritual code alongside a current ritual code, while mid/superior frontal regions showed this rotated code alongside a schema code. As outlined in the introduction, multiple animal[7] and human[4–6] studies have found rotated codes for sensory versus memory representations, which are thought to be a way to minimize confusion between information corresponding to currently perceived stimuli and information being held in working memory. The two previous fMRI studies that observed rotated codes used working memory tasks with simple stimuli: objects[5] and gratings[6]; the rotated codes in these studies were observed in posterior fusiform and visual cortex, respectively. Our study is the first to demonstrate rotated codes for temporally extended audiovisual narrative stimuli—here, the rotated codes were localized to medial occipital cortex and to mid and superior frontal regions. The difference in localization of the rotated codes between our study and previous studies likely relates to our use of more complex stimuli.

**Path code**. We also found a path code, separately representing each of the four paths (or sequences) present in our wedding video stimuli. Our results showed this path code most strongly in postcentral gyrus. This might be related to the fact that all these paths involve distinct sequences of actions, and research shows the involvement of the postcentral gyrus in representing and planning ahead of action sequences[66].

Previous multivariate fMRI studies of sequence coding have found that representation of predictable sequences that are context-dependent is localized to the hippocampus[8,9]; specifically, these studies found that hippocampus assigns distinct neural codes to identical items that are part of distinct sequences. Relatedly, studies using statistical learning and associative inference paradigms have also demonstrated that hippocampus plays a key role in integrating items that share common associates or appear adjacently in space or time (e.g., refs. 67–73; for a computational model of this process, see ref. 74). However, the hippocampus was not among the regions that passed FDR correction for the path code in our study. Notably, the aforementioned studies that found sequence coding in the hippocampus with fMRI did not have a hierarchical structure like ours did (i.e., where sequences were nested within the North / South schemas). It is possible that the hippocampus might preferentially represent the highest possible level of disambiguating structure in the environment (here, North / South); alternatively, hippocampus might simultaneously represent multiple levels in the hierarchy[75] but the presence of higher-level codes might make it harder to detect the lower-level codes with fMRI.

**Other codes**. Studies using other paradigms have found non-rotated codes for preceding and upcoming stimuli. For example, Ezzyat and Davachi[76] found that neural patterns associated with previous stimuli persisted in non-rotated form in the LO region within events. Also, fMRI studies of navigation have found (non-rotated) prospective representations of goal states[77,78].

As described in the "Results" section, we tested for non-rotated preceding/upcoming event codes across the brain (using the opposite of the contrast that we used to test for rotated preceding event codes) and did not obtain any significant regions (see the distribution of log Bayes factors in Supplementary Fig. 4).

There are several differences between our study and other studies that have found non-rotated preceding/upcoming event codes. For example, there was no demand to predict based on past and current stimulus identity in Ezzyat and Davachi[76] and thus no need to simultaneously and accurately represent both the past and current stimulus, which may explain why they found a non-rotated preceding event code. Also, in contrast to studies of navigation that observed prospective codes[77,78], participants in our study were passively viewing sequences as opposed to actively navigating, and they also had no way to determine the "end state" of a wedding at its outset

(participants could only determine the identity of the final ritual once they saw the campfire or flower ritual); both of these factors could decrease participants' incentive to use prospective codes.

## Relation between schema code and episodic memory

As noted above, Masis-Obando, Norman, and Baldassano[15] found a within-subjects relationship between the strength of schema representation during encoding of narratives (both audiovisual movies and audio narratives) and subsequent memory for details from those narratives. Guo and Yang[79] found a similar within-subjects relationship using an object-location task, and Raykov et al.[80] found a between-subjects relationship between a neural correlate of schema strength at encoding and subsequent recognition memory of details from previously viewed television shows. We obtained a similar relationship between schema activation and subsequent memory here (within-subjects, in our case), using a very different paradigm and carefully matching for the content of events across schemas, thereby demonstrating the generality of this finding.

An important contribution of our study is that we were also able to assess the relationship between other kinds of history-dependent representations (rotated preceding-ritual codes, path codes) and subsequent memory for details—this allowed us to address the question of whether all history-dependent representations help to scaffold subsequent memory, or whether high-level schema representations are special in this regard. In keeping with this hypothesis, North/South schema representations were the only ones to significantly correlate with subsequent memory. However, at the same time, the correlations we observed for the schema code were not (for the most part) significantly larger than the correlations we observed for the other codes. As such, we are not in a position to say that the schema code is a reliably better predictor of subsequent memory, compared to the other codes. Importantly, even if a particular history-dependent code ends up not correlating with memory, it can still contribute to performance in this task in other ways, such as by disambiguating which event is coming next (e.g., if the current ritual is coin, knowing that the preceding ritual was campfire, or that you are on the campfire-coin-egg path, will both allow you to correctly predict that the next ritual will be egg).

## Limitations

Our experimental design had several limitations. As noted above, participants in our study were passively viewing sequences as opposed to actively navigating, which may have reduced participants' incentive to use prospective neural codes. Also, the amount of initial exposure to each of the schemas on Day 1 of the study was fairly low (there were just 12 examples of North and South weddings, respectively) and the interval between the two days of the study (approximately 24 h) did not leave much time for consolidation. Both of these factors may have limited the development of schema representations in mPFC (for evidence that mPFC schema codes may take time to consolidate, see, e.g., ref. 81). Another limitation is that all weddings followed the North or South schemas. It is possible that mPFC represents a higher-order schema that encompasses both North and South weddings; however, our study was not equipped to look at this—identifying this higher-order schema would require contrasting North/South weddings with other (non-schematic) weddings, which we did not include in our study.

In addition to the aforementioned limitations of our design, our analysis approach also had several limitations. The neural patterns that we used for the representational similarity analysis were time-averaged over the entire ritual, which might make the analysis insensitive to predictive or retrospective codes that were only active during part of the ritual. Another limitation, specifically relating to path codes, is that our analysis assumed that neural codes were completely stable within a path, whereas other work has shown that hippocampal codes for sequences drift within the sequence (e.g., ref. 8) in the manner predicted by temporal context models[82]; to the extent that this drift was present in our study, it would reduce our ability to identify the path code.

## Conclusion

In this study, we discovered that multiple history-dependent neural codes exist in the brain: an abstract schema code, a path (sequence) code, and a rotated preceding ritual code that represents current and preceding rituals in an anticorrelated fashion, likely to avoid interference. We also found that the strength of the neural schema code during viewing of a particular wedding correlated with subsequent memory for the details of that wedding; this provides converging neural support for the idea that schemas act to scaffold memory for unique episodic details.

## Data availability

The raw data are available on openneuro.org (https://openneuro.org/datasets/ds005050). Processed data is available on Zenodo (https://zenodo.org/records/14620539).

## Code availability

The code is available on Zenodo (https://zenodo.org/records/14620539). The video stimuli, written version of the audio from the stimuli, and the list of details used for manual scoring of the recall of these stimuli has been added to OSF.

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

## Acknowledgements

We thank Kalyana Duggal, Alexandra de Soares, Hope Kean and Olivia Reblando for help during stimulus development; Andre Beukers for devising the context-dependent learning structure used in this paper; Uri Hasson for valuable input on the experimental design; Jeongmin Cho for help with data analysis; Sam Nastase for advice on data visualization; and other members of the Hasson and Norman laboratories for their comments and support.

This work was supported by a Multi-University Research Initiative Grant (ONR/DoDN00014-17-1-2961) and an NWO Rubicon grant (446-17-009). The funders had no role in study design, data collection and analysis, decision to publish or preparation of the manuscript.

## Author contributions

Silvy H.P. Collin: Conceptualization, Software, Formal analysis, Investigation, Data curation, Writing—original draft, Visualization. Ross P. Kempner: Software, Formal analysis, Writing—review and editing, Visualization. Sunita Srivatsan: Software, Formal analysis. Kenneth A. Norman: Conceptualization, Supervision, Funding acquisition, Project administration, Writing—review and editing.

## Competing interests

The authors declare no competing interests.
