## [Transparent Peer Review file · Communications Psychology]

Neural codes track prior events in a narrative and predict subsequent memory for details

Corresponding Author: Dr Silvy Collin

Version 0:

Decision Letter:

Dear Dr Collin,

Your manuscript titled "Neural codes track prior events in a narrative and predict subsequent memory for details" has now been seen by two of the original three reviewers, whose comments appear below. In light of their advice I am delighted to say that we are happy, in principle, to publish a suitably revised version in Communications Psychology.

We therefore invite you to revise your paper one last time to address the remaining editorial concerns and requests. At the same time we ask that you edit your manuscript to comply with our format requirements and to maximise the accessibility and therefore the impact of your work.

EDITORIAL REQUESTS:

Your manuscript puts forward a claim of specificity that is not sufficiently supported by appropriate statistics. While you do provide positive evidence derived from NHST for the schema code, there is no positive evidence for the respective absence of any of the other types of code, which cannot rely on non-significant findings in NHST. By journal guidelines, you may not treat this as evidence for an absence of an effect, may not interpret the null findings (which is currently the case in Results and Discussion), nor claim specificity. In revision, please provide appropriate statistical evidence as explained on our author pages: <https://www.nature.com/commspsychol/submit/submission-guidelines#statistical-guidelines> and in the attached "Editorial Requests Table".

SUBMISSION INFORMATION:

OPEN ACCESS:

* TRANSPARENT PEER REVIEW: Communications Psychology uses a transparent peer review system. On author request, confidential information and data can be removed from the published reviewer reports and rebuttal letters prior to publication. If you are concerned about the release of confidential data, please let us know specifically what information you would like to have removed. Please note that we cannot incorporate redactions for any other reasons.

Link Redacted

Best regards,

Troy Lui

Troy Lui, PhD
Associate Editor
Communications Psychology

REVIEWERS' COMMENTS:

Reviewer #1 (Remarks to the Author):

The authors have done a good job at addressing all of my concerns. This is a novel and interesting paper that makes important contributions to our understanding of how schemas support memory.

Reviewer #2 (Remarks to the Author):

The authors have addressed my concerns, and I believe this paper is now ready for publication.

Editorial Office

RE: Manuscript for consideration of publication in Communications Psychology.

Dear editorial office,

We would like to thank you for the opportunity to transfer a revised version of our manuscript to Communications Psychology.

We are grateful to the reviewers for their thoughtful and constructive comments on our work. Below, we have included a point-by-point response to each of the reviewers' comments. We would also like to highlight two major changes that we made in response to the reviews:

- 1) We have clarified how our results support the existence of a rotated preceding ritual code, as distinct from the current ritual code. As part of this, we now separately report regions that show *both* a rotated preceding ritual code and a current ritual code, as well as regions that show a rotated preceding ritual code *but not* a current ritual code, and regions that show a current ritual code *but not* a rotated preceding ritual code.
- 2) We decided to remove the k-means analyses from the paper. While we are fond of these analyses, it is apparent upon reflection that they do not lead to any conclusions about neural coding in this task that were not already supported (in a more direct and interpretable way) by the RSA analyses. Furthermore, the k-means analyses were nonstandard and somewhat complicated. We now intend to report these results in a separate, methods-focused publication that gives us more room to describe this novel analysis approach. If the reviewers feel strongly that we should report the k-means results in this paper, we can certainly add them back in as a supplement.

In addition to responding to the reviewers' comments, we made the raw data available on openneuro (see <https://openneuro.org/datasets/ds005050>) and the code and processed data available via GitHub (see https://github.com/silcol/publicRepo_Collinetal2024), and we determined effect sizes for all statistical tests performed.

We think that the paper has been substantially improved as a result of the feedback that we received, and we hope that this revised version is suitable for publication in Communications Psychology.

Sincerely,

Silvy Collin, Ross Kempner, Sunita Srivatsan and Kenneth Norman.

Response to reviewers' comments:

Reviewer #1 (Remarks to the Author):

Re: Collin et al., "Neural codes track prior events in a narrative and predict subsequent memory for details"

This is an interesting study of how different elements of event sequences are represented in the brain. In particular, the study focusses on the role of schemas in supporting our representation and memory for complex events. There has currently been a large amount of interest in these questions and this has generated a lot of interesting new findings. One limitation to most studies of schemas that use complex naturalistic tasks is that they rely on real-world “pre-experimental” knowledge, and it is difficult to control or manipulate such knowledge. The key strength of the present study is in its experimental procedure; it uses custom-made animations of fictional wedding rituals from two different “cultures”. Consequently, schematic knowledge about the events is tightly controlled and the transitions between events in the sequence of the “wedding” conform to rules created by the experimenters. This is an impressive achievement.

The authors argue that their procedure enables them to show evidence for parallel “distinct” representation of different types of schematic information about the events. I am not convinced that the evidence fully supports this claim, for reasons I give below. The second claim is that neural schema representations scaffold memory for specific details. This is supported by the data in the manuscript, though previous studies have also found similar results.

Specific points:

Comment 1:

The authors use RSA to identify four different “neural codes”, as shown in Figure 4. The “schema” code, which codes for whether a particular wedding is from the North or South culture, appears consistent with previous studies which have reported “storyline-specific” (Chang et al., 2021) or “script-specific” (Baldassano et al., 2018) neural representations. They also overlap with numerous studies that have reported neural representations of specific events. I appreciate that the “culture” manipulation here is somewhat different from these other examples, but they seem broadly similar.

The schema code in our study is indeed similar to the schema codes reported in these (and various other) studies to which we refer in the discussion. We now address Chang et al. (2021) in the discussion, since we agree with the reviewer that our schema effect is similar to their storyline-specific effect. However, a key difference between our study and these other studies mentioned by the reviewer (e.g., Chang et al., 2021) is that the other studies have content differences between schemas / storylines, making it harder to examine whether neural differences (also) reflect coding of current content versus past content. Among these studies, ours is unique with regard to the fact that each ritual is equally-often part of each schema, which allows us to conclude that the schema code is history-dependent. We have revised the discussion to highlight this point. Another novel feature of our study is that it simultaneously identifies multiple history-dependent codes (as distinct from current event codes) and separately assesses how each of these codes relates to subsequent memory for details.

The relevant paragraph of the discussion now reads as follows:

“As noted above, we found a schema code in a wide range of regions. The most comparable study to ours is Baldassano et al (2018), which also looked at schema representations for temporally extended narratives.

However, the nature of schemas they (and other schema/narrative studies, e.g., Chang et al., 2021) investigated was different for several reasons. First, we used a novel schema (devised by us), whereas Baldassano et al (2018) examined schema representations for commonly-experienced events like going through airport security and ordering food at a restaurant (see also Masis-Obando et al., 2022, which reported analyses of the memory retrieval data from this study). Second, these earlier studies (Baldassano et al., 2018; Chang et al., 2021) both have content differences across the schemas, while in our design each ritual is equally often part of each schema. This makes it possible to investigate schema differences without possible confounds due to content differences. These differences between the task designs notwithstanding, it is still useful to compare our results to theirs.”

And we also now relate our schema code results to the results from Chang et al. (2021):
“We indeed found a robust schema code in the thalamus and basal ganglia regions (pallidum, caudate). This is in line with earlier research (Chang et al., 2021) that found storyline-specific effects in these subcortical areas.”

Comment 2:

The authors also identify a current ritual code, mostly in the occipital cortex. This is readily interpretable, as it is easy to imagine that the rituals contain enough specific information to distinguish the representation of each one from the others. The authors also report evidence for regions that represent the previous ritual – specifically that the representation of the current ritual is anti-correlated to the representation of the preceding ritual. This is an interesting novel finding, and the authors cite studies from the working memory literature which have described similar effects. However, I wonder how much this effect is driven by the “current ritual” code? Comparing the plots in Fig 7 to the upper lefthand four plots in Supplementary Figure A3 (“Posthoc confirmation of the pattern for the current ritual neural code”), the violin plots look highly similar – though they are argued to be showing evidence for different kinds of representations. Apologies if I am misreading these plots. My broader concern is that the maps for the rotated preceding ritual and the current ritual are overlapping and the contrasts used to identify them seem to be similar. Also the “rotated preceding ritual code” regions identified by the K-Means analysis is largely the same area identified as supporting “current ritual” representations from the RSA searchlight analysis. Presumably the regions argued to support rotated preceding rituals don’t show significantly more evidence for these representations than the current ritual? If so, can the authors confidently rule out the explanation that there is only strong evidence for representations of the current ritual?

We realize that the rotated preceding ritual code is complex to address and apologize for the confusion. However, we do believe that our results show convincing evidence for a rotated preceding ritual code, and hope that the changes listed in this answer help to convey this result better to the reader.

Importantly, the rotated preceding ritual code involves a comparison across stages (different from the current ritual code). In other words, as we state in the paper, the RSA contrast for the rotated preceding rituals code looks for regions where patterns during the coin/torch stage were especially dissimilar to the template corresponding to the preceding ritual. For example, if the coin stage were preceded by the campfire stage, we would expect it to be less similar to the campfire stage template relative to the flower stage template). Any region that was *only* sensitive to the current ritual (e.g., coin/torch) would, by definition, not be affected by whether the preceding ritual was campfire or flower, and it would therefore show a null effect for the preceding-ritual RSA contrast.

We believe that the understandable confusion relates to a lack of clarity on our part in describing the relationship between the preceding-ritual and current-ritual RSA contrasts. Whereas (as noted above) the preceding-ritual RSA contrast is defined by a comparison *across stages* (i.e., the upper-right and lower-left cells in Figure 7), the current-ritual RSA contrast is defined by a comparison *within stages* (i.e., the upper-left cell in Figure 7). Thus, the preceding-ritual and current-ritual contrasts are defined based on completely distinct comparisons. This point is also illustrated by the RSA matrices in Figure 11: The squares that are grayscale for the preceding ritual code are white for the current ritual code, meaning that they were omitted from the latter analysis, and the opposite is also true – the squares that are grayscale for the current ritual code are white for the preceding ritual code.

The fact that preceding-ritual and current-ritual contrasts are defined based on distinct comparisons means that it's possible for a particular region to show *both* a preceding-ritual and current-ritual code; it is also possible for a region to show a preceding-ritual code but not a current-ritual code, and for a region to show the opposite (a current-ritual code but not a preceding-ritual code). As it turns out, all three of these region types exist in our data. To clarify this, we have substantially revised how we present these results in Figure 4: Instead of plotting preceding-ritual and current-ritual regions separately (as we did before, leaving it to readers to notice areas that overlap and do not overlap), we now report separate brain maps for three non-overlapping sets of regions:

1. regions that pass FDR correction for both codes (i.e., the overlap in the old results), now shown in Figure 4C (blue color map)
2. regions showing a preceding-ritual code but not a current-ritual code (identified by taking the regions that pass FDR correction for preceding-ritual, and subtracting out regions that show current-ritual coding at a liberal threshold of $P < .1$ uncorrected), now shown in Figure 4C (red color map)
3. regions showing a current-ritual code but not a preceding-ritual code (identified by taking the regions that pass FDR correction for current-ritual, and subtracting out regions that show preceding-ritual coding at a liberal threshold of $P < .1$ uncorrected), now shown in Figure 4C (green color map)

Besides including these results in Figure 4, we also calculated the violin plots (and corresponding posthoc t tests) separately for these three regions, which are included in the corresponding figures. We also ran the correlation to behavior for all three of these neural codes (all non-significant). We are confident that this analysis will be less confusing than our original analysis.

Additionally, as noted at the start of the response letter, we have removed the k-means results from the manuscript. The results of $K = 6$ actually do represent the RSA results quite well (i.e., 5 clusters corresponding to the 5 codes reported in the revised paper + a null cluster). However, we believe that the revised RSA results convey the relevant information about neural codes in a clear and direct way, and that the k-means results a) are not needed to get across our main points and b) would only sow confusion due to their being a novel and non-standard method. Therefore, we now believe that the k-means results deserve their own paper focused on describing and evaluating this new analysis approach.

Comment 3:

The fourth neural code is for the “path” – the specific sequence of actions in the current ritual. There was just a small region in the postcentral gyrus that showed evidence for this representation (though the plot in Figure 6 suggests that this is not a particularly strong effect). This theoretical importance of this effect is difficult to ascertain; the authors do not discuss it at all, and only discuss the lack of an effect in the hippocampus.

The path code being present in the postcentral gyrus rather than hippocampus was indeed surprising. We believe that a reason might be that these rituals all involve distinct motor actions, and postcentral gyrus is known to be involved in representing and planning ahead of such sequences of actions (as e.g. shown in Gallivan et al, *Cerebral Cortex*, 2016). We now speculate on this reason for the path code to be present in the postcentral gyrus in the discussion.

The relevant part of the discussion now reads as follows:

“We also found a path code, separately representing each of the four paths (or sequences) present in our wedding video stimuli. Our results showed this path code most strongly in postcentral gyrus. This might be related to the fact that all these paths involve distinct sequences of actions, and research shows the involvement of the postcentral gyrus in representing and planning ahead of action sequences (Gallivan et al., 2016).”

Gallivan, J. P., Johnsrude, I. S., & Flanagan, J. R. (2016). Planning Ahead: Object-Directed Sequential Actions Decoded from Human Frontoparietal and Occipitotemporal Networks. *Cerebral cortex (New York, N.Y. : 1991)*, 26(2), 708–730.

<https://doi.org/10.1093/cercor/bhu302>

Comment 4:

Finally, the authors relate the strength of their schema effect with memory for details of the weddings. This is a nice result – and perhaps a clearer demonstration of this effect than in previous studies. Nevertheless, other studies have also shown relationships between activations of schemas and memory (notably the Masis-Obando et al. 2022 study discussed by the authors, but also a recent study by Guo and Yang [2023, *Cerebral Cortex*], and the study by Raykov et al [2021, *Cerebral Cortex*]).

In addition to the similarity to Masis-Obando (2022), we agree that our effect is indeed related to the effects found by Guo and Yang (2023) as well as Raykov et al. (2021), which are now also covered in the discussion. As noted above, an important difference between these studies and our study is that we defined schemas by a difference in transition structure while keeping the actual content the same; in this way, we control for the content of the events in our definition of schema. Therefore, our result expands the range of schema types whose neural activation at encoding has been shown to correlate with subsequent memory behavior.

The relevant part of the discussion now reads as follows:

“As noted above, Masis-Obando et al. (2022) found a within-subjects relationship between the strength of schema representation during encoding of narratives (both audiovisual movies and audio narratives) and subsequent memory for details from those narratives. Guo et al. (2023) found a similar within-subjects relationship using an object-location task, and Raykov et al. (2021) found a between-subjects relationship between a neural correlate of schema strength at encoding and subsequent recognition memory of details from

previously-viewed television shows. We obtained a similar relationship between schema activation and subsequent memory here (within-subjects, in our case), using a very different paradigm and carefully matching for the content of events across schemas, thereby demonstrating the generality of this finding."

Taken together, the findings are intriguing. Potentially the most important and novel result is the existence of "rotated previous ritual" codes. However, at the moment I am not convinced that there is strong evidence for this.

Reviewer #2 (Remarks to the Author):

The authors undertook a two-day fMRI study with the aim of characterizing the presence of history-dependent neural representations during a sequence learning task and the relationship of these neural codes with subsequent memory. On the first day the participants learned schemas as they watched animated videos of weddings that followed predictable sequences of rituals depending on if the couple was from the North or South of a fictional island. The next day, they watched new wedding videos in the scanner, followed by a free recall test and schema test. The weddings were structured with "rituals" in predictable sequences, which allowed them to examine schema effects (neural codes associated with North vs South weddings), path effects (codes for weddings following a particular sequence within each schema), and rotating code effects (whether the preceding ritual was represented at the same time as the current one in an orthogonal manner). Using a searchlight RSA approach, they found evidence of schema coding in various cortical and subcortical regions, path coding in the precentral gyrus, and rotated ritual coding in occipital cortex, thus showing that different brain regions code different kinds of history-dependent information. They found the "strength" of neural representation of the North/South schema for a given wedding was associated with subsequent memory for the details of that wedding, suggesting that schemas provide a scaffold for episodic memory details within-subjects.

Overall, I thought this was an interesting and well-designed experiment. The question of how we use previously experienced information in a given moment has been historically difficult to test, and I think will be of interest to the readership of this journal. The paper was well-written, the authors tested a large sample of participants, and their models made sense within the context of their design.

I have some questions about the methods, which were unclear at times, as well as the interpretation.

Comment 1:

1. Are the dots in figure 3b individual participants? It looks like some of them didn't learn the schema very well. Were there any exclusion criteria for learning, and was anyone at chance?

Yes, those dots are individual participants. There are indeed differences across participants in terms of schema learning, although in general the schema learning performance was good. We decided against removing participants based on their performance on this schema test, and only exclude participants in case they performed very poorly on the stop-and-ask attention questions (which would be evidence for this participant simply not paying attention during the task). The reason for not excluding based on schema test at the end of day 2 is

that the few participants who performed poorly on this test did not necessarily do badly during the video viewing itself. For example, one of the poorly performing participants in the schema test was correct only 29.13% of the time (responding 14.54% with the incorrect ritual from the correct stage, and 14.08% on average for each of the four options from the wrong stage). However, this person performed much better for the stop-and-ask prediction questions during the video viewing itself (97.2 % correct). Thus, while this particular person might not have shown particularly good schema knowledge during the final schema test, he/she did learn the schema during viewing. For that reason, it did not seem correct to remove this person from the dataset, and we decided to be conservative in removing participants and include everyone.

Comment 2:

2. What is the reasoning for averaging across TRs for the searchlight but including the TR information in the k means clustering, was this a methodological or theoretical decision? The k means clustering identified many more regions for path coding than the searchlight. How should the reader interpret these differences? Is it that temporal information across TRs are more informative, and does this tell us something about how path information is represented?

We included the TR information in the k-means clustering analysis because we wanted to make as few assumptions as possible about the underlying codes. That is an intriguing hypothesis about temporal information being important for the path code, but there is a much simpler explanation for the discrepancy between the path regions obtained by the RSA searchlight vs. k-means, namely that a much more stringent criterion was applied in the former case: to appear in the RSA searchlight, a region had to pass FDR multiple-comparisons correction, whereas there was not *any* statistical threshold that had to be passed in order for a region to appear in the k-means path cluster (“path cluster” was just our post-hoc way of describing that cluster). When we ran statistical tests on the k-means cluster, the statistical support for this region truly being a path cluster was weak (as shown in Figures A8 and A9 from the original manuscript). If it had been stronger, the region would have shown up in the RSA analysis.

As noted at the start of the response letter, we have removed the k-means results from the manuscript. The results of $K = 6$ actually do represent the RSA results quite well (i.e., 5 clusters corresponding to the 5 codes reported in the revised paper + a null cluster). However, we believe that the revised RSA results convey the relevant information about neural codes in a clear and direct way, and that the k-means results a) are not needed to get across our main points and b) would only sow confusion due to their being a novel and non-standard method. Therefore, we now believe that the k-means results deserve their own paper focused on describing and evaluating this new analysis approach.

Comment 3:

3. The null cluster consists of regions the literature would typically conceive of as related to schematic or semantic processing, so it's a little surprising that these are the only areas with a null effect. Perhaps there is a higher order schema extracted across both North and South weddings for the rituals themselves, like campfires at a wedding, that is not dependent on a sequence (i.e. not a ritual specific to a path and schema, but a representation extracted across paths and North/South schemas). Indeed, it is feasible that participants extract such

information after seeing so many slightly different instances of people having campfires at a wedding, for example, regardless of where that campfire is in the sequence. Much schema research on the mPFC is not sequence related, so perhaps basing your schemas on the ordering of events would not target this region. Could this explain the null results in the mPFC/ATL regions? Or could it be a tSNR issue?

As discussed in the discussion 3rd paragraph of section 3.2.1, we also found it surprising to not have a schema code in the mPFC, as was shown in many studies before. We speculated on why this is the case, for example because of insufficient initial exposure to the schemas or because of the interval between learning and viewing novel weddings being short. However, the explanation given by the reviewer above is also a plausible one, and we added it to this section in the discussion.

The relevant part of the discussion reads as follows:

“... ; yet another possibility is that mPFC might encode a higher-order schema that encompasses both the North and South types of weddings.”

Comment 4:

4. I don't think the authors operationalize “neural strength”. What exactly are you correlating with behavior?

The strength of a particular neural code is computed by running the RSA contrast for that code (as described in the “Representational Similarity Analysis” subsection of the Methods) for that particular wedding. We now explain this in the Methods.

The relevant part of the Methods (in subsection called “Relating brain to behavior”) now reads as follows:

“We related the brain results to individual participants' behavior. First, for each wedding within each participant, we computed the “neural strength” of each code by running the RSA contrast for that code (see “Representational Similarity Analysis” section above) on the data from that wedding.”

Comment 5:

5. The authors discuss possibilities as to why they did not observe the hippocampus as a region representing the path code, but why do the authors think the postcentral gyrus was identified? Why do they think the insula is involved in the schema code?

The path code being present in the postcentral gyrus rather than hippocampus was indeed surprising. We believe that a reason might be that these rituals all involve distinct motor actions, and postcentral gyrus is known to be involved in represented and planning ahead of such sequences of actions (as e.g. shown in Gallivan et al, Cerebral Cortex, 2016). We now speculate on this reason for the path code to be present in the postcentral gyrus in the discussion.

The relevant part of the discussion now reads as follows:

“We also found a path code, separately representing each of the four paths (or sequences) present in our wedding video stimuli. Our results showed this path code most strongly in postcentral gyrus. This might be related to the fact that all these paths involve distinct sequences of actions, and research shows the involvement of the postcentral gyrus in representing and planning ahead of action sequences (Gallivan et al., 2016).”

Gallivan, J. P., Johnsrude, I. S., & Flanagan, J. R. (2016). Planning Ahead: Object-Directed Sequential Actions Decoded from Human Frontoparietal and Occipitotemporal Networks. *Cerebral cortex*, 26(2), 708–730. <https://doi.org/10.1093/cercor/bhu302>

The insula being involved in the schema code is also surprising to us – we are reluctant to speculate here (no good ideas come to mind) but we now note in the paper that other studies have found that the insula is involved in representing event schemas (Baldassano et al., 2018) and in anticipatory coding of upcoming events (Lee, Aly, & Baldassano, 2021).

The relevant part of the discussion now reads as follows:

“Surprisingly, as outlined above, we also found the schema neural code in the insula. We do not at present have a good account of what role the insula is playing here, but we do note that other studies have found that the insula is involved in representing event schemas (Baldassano et al., 2018) and in anticipatory coding of upcoming events (Lee, Aly, & Baldassano, 2021).”

Comment 6:

6. Can the authors speculate as to what the functional relevance is of these different kinds of neural codes? Only one seems important for subsequent memory, why are the others important for cognition?

Only one seems important for scaffolding memory for episodic details; however, the other codes can still contribute to prediction of upcoming events, by disambiguating which event is coming next (e.g., if the current ritual is coin, knowing that the preceding ritual was campfire, or that you are on the campfire-coin-egg path, will both allow you to correctly predict that the next ritual will be egg). We now address these points more explicitly in the discussion, with the final part of relevant discussion paragraph now reading as follows:

“Thus, only the schema code turned out important for scaffolding memory for details; however, the other history-dependent codes can still contribute to performance in this task by disambiguating which event is coming next (e.g., if the current ritual is coin, knowing that the preceding ritual was campfire, or that you are on the campfire-coin-egg path, will both allow you to correctly predict that the next ritual will be egg).”

Minor

Comment 7:

- Adding a label for the stages in Figure 1B may be helpful since you often refer to them as stages, and I wasn't initially sure if the “start” was stage 1 or if stage 1 started at campfire.

Thanks for raising this point, we added labels to Figure 1B now.

Comment 8:

- What similarity metric was used for the searchlight, was it a Pearson's correlation? Was it fisher transformed?

Yes, it is Fisher transformed and was Pearson's correlation. This information is now added to the methods (similarity calculated with numpy corrcoef function in Python).

Comment 9:

- It would be helpful to include standard error bars on your bar charts

Thanks for raising this point, we added those now (see Fig 3B and 10).

Comment 10:

- Are these analyses in volume space or surface space? The functional preprocessing section suggests surface space, but you also have subcortical findings so I assume it was done in volume space? If the surface space wasn't used for these analyses it might be best to remove this to avoid confusion.

Thanks for raising this point; these analyses are in volume space and we have now made this clear by removing the surface space text from the functional preprocessing section.

Comment 11:

- You mention the functional data were high pass filtered at 128s in the functional section but 480s in the 5.5.3 section.

Thanks for highlighting this confusing point in the methods; 480 sec was used for high pass filtering, and we have now corrected this in the methods.

Comment 12:

- Were there any exclusion criteria for motion?

Someone with really excessive head motion would have been excluded from the analysis (but this was not the case for anyone).

Reviewer #3 (Remarks to the Author):

Collin et al present a two-day fMRI study that investigates how context-dependent predictions about the future are made and represented in the human brain. The authors describe an interesting behavioral task in which participants learned sequences of wedding ceremonies for two hypothetical cultures. The authors manipulated the transition structure between each component of the wedding rituals, and participants learned the transition sequences on Day 1. This allowed the authors to compare neural activity recorded during viewing of the rituals and to contrast three different models of the representation of predictions based on schematic knowledge. On Day 2, the authors also tested memory for details from specific wedding ceremonies that followed the broader schematic structure. The authors show that different levels of representation exists across a hierarchy of cortical and subcortical areas and demonstrate that these representations relate to memory.

This study uses an interesting design, and asks important questions about memory, however there are significant limitations in terms of the predictions, alternative hypotheses, and the correlation between the neural schema representations and memory performance.

Major Comments**Comment 1:**

(1) Three types of predictive knowledge: The authors describe three ways that a person could generate a prediction about the immediate future, given information about the present. The first two models are (1) a specific sequence of future actions; (2) a general schema. I

was confused about the meaning of the third model. The way it is described (“carry forward a representation of the scene”) makes it sound like there is no prediction happening. I interpreted this as follows: this kind of “absence of prediction” could be useful in situations in which there is not enough information to generate an informed prediction about the immediate future (e.g. it is not even possible to identify the space of possible future states). But an alternative situation in which the “carry forward” model could be useful is when there are multiple identifiable future states and there is high uncertainty about which one or ones are most likely (e.g. a state of high entropy). These two interpretations seems slightly different—are either what the authors intended? Complicating things further, the example given in the Introduction doesn’t exactly fit with either (you see someone outside their apartment and, knowing they were just inside of it, predict the person is leaving). I’m just confused about what kind of situations this third “carry forward” model was supposed to account for.

In our view, all of the different kinds of history-dependent codes are useful for the purpose of making predictions, including a “carry forward code”, as described by the reviewer. The results show strong evidence that a rotated version of the stage 2 ritual code is present during stage 3 (i.e., a “retrospective code”), but we did not observe evidence for prospective coding in this study (see discussion section “codes we did not observe”). In other words, there was no evidence for stage 3 code (rotated or non-rotated) during stage 2. However, importantly, the absence of neural evidence for predictions does not necessarily mean that participants were not making predictions during the task. We expect it to be possible that moments of prediction might have been too brief and/or idiosyncratically timed, which could be why it does not appear in our fMRI analyses. Most studies that have obtained fMRI evidence for prediction have used shorter stimulus durations. We have extended our treatment of these points (see section “non-rotated previous and upcoming ritual codes”) in the discussion section, which reads as follows:

“... Besides that, the moments during which participants were making predictions might have been too brief in our task design and/or idiosyncratically timed, which could be yet another reason why we did not find a neural code for upcoming events.”

Comment 2:

(2) Specificity of rotated representations: Related to the previous point, reading about the third “carry forward” model made me think about interference, and indeed the authors bring up the issue of relevant vs. irrelevant stimuli in the context of the working memory studies they cite in the Introduction. Why wouldn’t such interference be an issue for the schematic and path models, in addition to the carry forward model?

We do believe one would need rotated ritual codes, but not necessarily rotated schema and path codes because, if you simultaneously represent two rituals – the previous one and the current one – you need some way of keeping track of which one is previous and which one is current. For schema codes and path codes, there is no need to represent two instances of the same type of representation at the same time. In other words, it is potentially useful to represent the previous and current rituals (since that helps specify the next one), but it is not potentially useful to actively represent the north and south schemas at the same time, and it is not potentially useful to represent two paths at the same time. It might happen that a person is confused and they represent two paths because they are unsure of which one they are on, but there is no benefit to simultaneously representing two paths and hence there is

no need to represent them in a rotated form. We made some changes to the introduction in order to highlight this explanation in the manuscript. The preceding ritual code explanation in the introduction now ends with this text:

“We do not expect such anticorrelated patterns for schema or path codes, since, unlike for rituals, there is no need to represent multiple schemas or paths at the same time in order to accurately predict upcoming events.”

Additionally, in reference to the role these codes play in reducing interference in general, we agree that all of these codes play an important role in reducing interference. The more strongly you represent the correct schema (but also path, current ritual, previous ritual), the better you will be able to anticipate what will happen next in a wedding.

Comment 3:

(3) Anatomically constrained predictions: The authors make no predictions about what brain areas might be important for representing the different forms of predictive knowledge about schemas, or whether the different representations would exist in different brain areas. This is important because demonstrating that the brain has the three types of representation is not necessarily informative—wouldn't you expect all three to exist, depending on the brain area and how you train people on learning the schema sequences?

We expected all three to exist at the same time, but in different brain regions. We expected schematic representations in the network of regions identified by Baldassano et al. (2018), path representations to be present in the hippocampus, and current ritual as well as rotated preceding rituals representations both to be present in visual cortex. In the previous version of the manuscript, we talked about these other studies in the discussion section (when describing how our results relate to prior work). In the revised manuscript, we have added text to the introduction to clarify what expectations we had about brain regions prior to running the study.

See this introduction text for schema code explanation:

“We predicted this schema code to be present in a network of brain regions identified as schema-sensitive regions in a similar study by Baldassano et al. (2018).”

And here for path code explanation:

“We predicted this path code to be present in the hippocampus, since the hippocampus is known to represent predictable sequences (Clarke et al., 2022, Hsieh et al., 2014).”

And here for previous ritual code explanation:

“Based on these findings, we expected that visual cortex representations of previous rituals, if present, would be anticorrelated with the patterns that are evoked by these same rituals when they are currently happening.”

Comment 4:

(4) RSA and K-means (Figure 4): In the searchlight analysis, the authors find almost no brain regions that represent the rituals using a “path” code, however in the K-means analysis there are huge areas of the posterior and medial parietal, and lateral temporal lobes that show a path code. For the other analyses (schema/current/rotated) this kind of discrepancy doesn't seem to exist—why would this be the case?

To appear in the RSA searchlight brain map, a region had to pass FDR multiple-comparisons correction, whereas there was not *any* statistical criterion that had to be passed in order for a

region to appear in the k-means path cluster (“path cluster” was just our post-hoc way of describing that cluster). When we ran statistical tests on the k-means cluster, the statistical support for this region truly being a path cluster was weak (see figures A8 and A9). If it had been stronger, the region would have shown up in the RSA analysis.

As noted at the start of the response letter, we have removed the k-means results from the manuscript. The results of $K = 6$ actually do represent the RSA results quite well (i.e., 5 clusters corresponding to the 5 codes reported in the revised paper + a null cluster). However, we believe that the revised RSA results convey the relevant information about neural codes in a clear and direct way, and that the k-means results a) are not needed to get across our main points and b) would only sow confusion due to their being a novel and non-standard method. Therefore, we now believe that the k-means results deserve their own paper focused on describing and evaluating this new analysis approach.

Comment 5:

(5) Analysis of memory: in the Methods, the authors describe two measures of memory: (1) number of episodic details and (2) number of rituals. For both, the authors took the difference between correctly and incorrectly remembered items. What do the authors mean by “incorrect”? Does this include omitted details/rituals? Does this also include falsely remembered details/rituals?

Incorrect here means that a participant explicitly mentions something that relates to another wedding. For example, a ritual that was not performed in that wedding or an episodic detail that did not belong to that wedding. This is now explained in the methods, as follows (in relating brain to behavior section):

“Incorrect details/rituals here refers to a participant explicitly mentioning a detail/ritual that relates to another wedding – for example, a ritual that was not performed at that wedding or an episodic detail that did not belong to that wedding (for a list, see Appendix).”

Comment 6:

(6) Memory–schema correlation (page 14): the Abstract and Introduction describe one of the major findings as being that neural representations of schemas are related to episodic memory. There are two issues with this finding. First, while it is true that the neural schema–memory correlation was significant for both types of memory test (details and rituals), these correlations were not significantly larger than those for any other pair of neural pattern and memory. The only exception is neural schema and memory for rituals, and even this was only significantly larger than the one for the neural code for the current ritual. Put another way, the neural schema representation performed better at predicting memory than a representation that completely ignores anything about the larger structure of the learned sequence (i.e. one that has no history). Second, this result was found when measuring memory for the rituals but NOT for the (episodic) details. In other words, neural schema representations were correlated with memory for information that is basically schematic. This seems to undermine the argument (made initially in the manuscript) that neural schema representations help scaffold episodic memory, which is supposedly a major finding.

We acknowledge that the schema-behavior correlations are not significantly larger than correlations between other neural codes and behavior. However, based on our results, one

can conclude that the strength of schema representations at encoding is related to subsequent memory, both for details and for rituals. This findings adds to existing findings (from e.g. Masis-Obando et al, 2022) by showing this for a different type of schema. We defined schema by a difference in transition structure while keeping the actual content the same. In this way, we control for the content of the events in our definition of schema. Therefore, our result is a conceptual replication and extension of the existing finding by expanding the range of schema types whose neural representations have been shown to correlate with behavior.

We agree that we are not justified in saying that schemas are more predictive of subsequent memory than other kinds of history-dependent codes, but we were careful to avoid saying this in the paper. For example, the discussion reads “However, at the same time, the correlations we observed for the schema code were not (for the most part) significantly larger than the correlations we observed for the other codes. As such, we are not in a position to say that the schema code is a reliably better predictor of subsequent memory, compared to the other codes”.

Minor Comments

Comment 7:

(1) Figures 5-8: it would be helpful to indicate visually which conditions differed from which other ones (e.g. comparison bars and asterisks).

Thanks for raising this point, we now included asterisks to indicate significance.